# FLASH-MONO: FEED-FORWARD ACCELERATED GAUSSIAN SPLATTING MONOCULAR SLAM

**Zicheng Zhang**[1], **Ke Wu**[1], **Xiangting Meng**[2], **Keyu Liu**[3], **Jieru Zhao**[3*], **Wenchao Ding**[1*]

[1]Fudan University    [2]ShanghaiTech University    [3]Shanghai Jiao Tong University

https://victkk.github.io/flash-mono

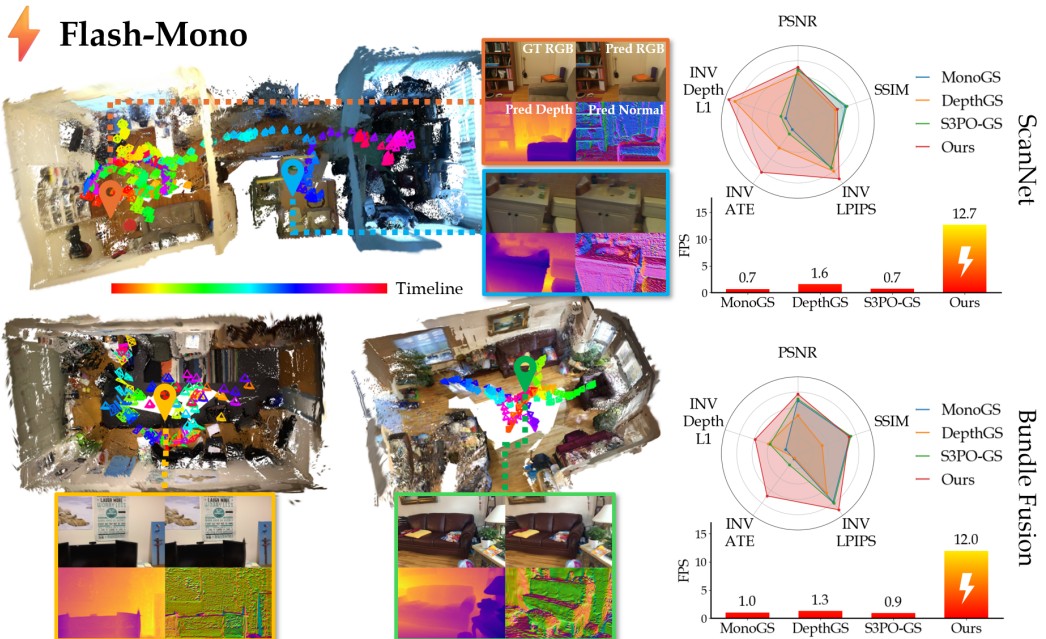

Figure 1: **Our Results for Reconstruction and Rendering & Tracking & Speed Metrics.**
Our method reconstructs high-quality Gaussian maps in complex scenes with multiple rooms and varying lighting conditions. The right-side radar chart shows our rendering quality (PSNR, SSIM, LPIPS) and trajectory tracking accuracy (ATE), with reciprocals of LPIPS, ATE, and Depth L1 plotted for clarity. Our method outperforms others in both rendering quality and trajectory accuracy, offering a **10x** speedup over contemporary monocular GS-SLAM methods.

## ABSTRACT

Monocular Gaussian Splatting SLAM suffers from critical limitations in time efficiency, geometric accuracy, and multi-view consistency. These issues stem from the time-consuming *Train-from-Scratch* optimization and the lack of inter-frame scale consistency from single-frame geometry priors. We contend that a feed-forward paradigm, leveraging multi-frame context to predict Gaussian attributes directly, is crucial for addressing these challenges. We present Flash-Mono, a system composed of three core modules: a feed-forward prediction frontend, a 2D Gaussian Splatting mapping backend, and an efficient hidden-state-based loop closure module. We trained a recurrent feed-forward frontend model that progressively aggregates multi-frame visual features into a hidden state via cross attention and jointly predicts camera poses and per-pixel Gaussian properties. By directly predicting Gaussian attributes, our method bypasses the burdensome per-frame optimization required in optimization-based GS-SLAM, achieving a **10x** speedup while ensuring high-quality rendering. The power of our recurrent architecture extends beyond efficient prediction. The hidden states act as compact submap descriptors, facilitating efficient loop closure and global $\mathrm{Sim}(3)$ optimization to mitigate the long-standing challenge of drift. For enhanced geometric fidelity, we replace conventional 3D Gaussian ellipsoids with 2D Gaussian surfels. Extensive

---

*Corresponding author.

experiments demonstrate that Flash-Mono achieves state-of-the-art performance in both tracking and mapping quality, highlighting its potential for embodied perception and real-time reconstruction applications.

# 1 INTRODUCTION

Recent advancements in real-time 3D scene reconstruction using a single RGB camera have attracted considerable attention. Its ability to provide dense and information-rich maps is crucial for applications ranging from robotic navigation and spatial intelligence. While traditional representations like point clouds, voxels, and surfels have been widely used, 3D Gaussian Splatting (3DGS) (Kerbl et al., 2023) has recently emerged as a highly promising approach for 3D reconstruction, owing to its capabilities in differentiable rendering, self-supervised training from RGB images, and high-fidelity novel view synthesis. Consequently, integrating 3DGS into a real-time monocular SLAM framework presents a significant opportunity for advancing embodied perception.

An early attempt at monocular GS-SLAM (Matsuki et al., 2024) initializes Gaussians randomly and relies on hundreds of optimization iterations per frame to maintain a consistent map. Subsequent methods (Zheng et al., 2025; Wu et al., 2025a) employ depth or optical flow prediction networks to provide geometric priors, which are used to initialize Gaussian geometric attributes. However, their performance remains limited to around 1 FPS, insufficient for real-time SLAM, as they do not abandon the *Train-from-Scratch* paradigm (Gaussian appearance attributes are randomly initialized and trained). Moreover, these approaches also suffer from severe multi-view inconsistencies, as monocular depth predictions are inherently scale-inconsistent. On a different front, feed-forward methods such as VGGT (Wang et al., 2025a) have demonstrated excellent multi-frame consistency by applying cross-attention over image batches. While the feed-forward approach supplies a consistent geometric prior, its offline requirement of processing all frames at once makes it fundamentally incompatible with the streaming input and low-latency pose estimation required by SLAM.

Based on this analysis, we identify three critical challenges that impede the development of a truly real-time and globally consistent monocular GS-SLAM system. First, the prevalent *Train-from-Scratch* paradigm of Gaussian Splatting requires dozens to hundreds of iterations of optimization per keyframe, fundamentally preventing real-time performance. Second, incremental feed-forward reconstruction methods are susceptible to cumulative pose and scale drift, as past predictions cannot be refined by future observations, leading to poor multi-frame geometric consistency. Third, vanilla 3DGS representations often suffer from poor geometry quality.

To overcome these challenges, we propose **Flash-Mono**, a monocular GS-SLAM system designed to deliver exceptional speed performance and high-quality mapping. At its core is a recurrent feed-forward reconstruction model that incrementally predicts camera poses together with a dense, pixel-aligned Gaussian representation for each incoming frame. This design directly addresses the efficiency bottleneck of optimization-based GS-SLAM: instead of training Gaussians from scratch at every keyframe, we predict high-quality Gaussians and only apply lightweight backend refinement. To combat the drift that is common in incremental feed-forward reconstruction, we leverage the model's hidden state as a compact submap descriptor: when revisiting a location, a single conditional forward pass produces an accurate $\mathrm{Sim}(3)$ loop constraint, which we integrate into pose graph optimization for global correction. Finally, to improve geometric fidelity, we adopt 2D Gaussian surfels as our map primitive, providing a stronger surface prior than vanilla 3DGS. With these components, Flash-Mono supports streaming inputs while achieving real-time performance and globally consistent reconstructions. In summary, our main contributions are:

- We propose a real-time (10 FPS+) monocular GS-SLAM framework that leverages a recurrent feed-forward model to predict poses and Gaussians directly. Compared to all previous methods that require training Gaussians entirely from scratch, our framework achieves remarkable speed improvements while still ensuring high-quality results.
- We design a novel and efficient loop closure method based on the hidden state of the feed-forward model, and through $\mathrm{Sim}(3)$ graph optimization, we mitigate accumulated errors while preserving the global consistency of the reconstructed map.
- We conduct extensive experiments on large-scale and challenging datasets, evaluating rendering, geometry, tracking, and efficiency metrics. Our work achieves state-of-the-art re-

sults in both tracking and rendering quality, while significantly surpassing previous methods in processing speed.

## 2 RELATED WORKS

**SLAM with 3D Foundation Model.** Feed-forward architectures have recently emerged as a powerful alternative to classical Structure-from-Motion (SfM) pipelines, which rely on iterative feature matching and bundle adjustment (Schönberger & Frahm, 2016). Early works such as DUSt3R (Wang et al., 2024) and its extension MASt3R (Murai et al., 2025) pioneered this paradigm by directly predicting point maps from image pairs within a single forward pass. To overcome the limitation of pairwise inputs, Fast3R (Yang et al., 2025) introduced a transformer-based design capable of processing multiple images in parallel, thereby accelerating large-scale 3D reconstruction. CUT3R (Wang et al., 2025b) further advanced this direction by adopting a recurrent framework that accommodates a variable number of images and supports diverse input modalities, enabling online processing of video streams. Extending this line of work, VGGT (Wang et al., 2025a) demonstrated the potential of large-scale, multi-task learning for feed-forward reconstruction, while FLARE (Zhang et al., 2025) and Splatt3r (Smart et al., 2024) extended the idea to renderable Gaussian Splatting representations directly from unposed images.

Nevertheless, directly applying feed-forward methods to SLAM remains highly challenging due to the need for accurate pose consistency, temporal stability, and long-horizon robustness. For example, although MASt3R-SLAM (Murai et al., 2025) partially mitigates some of these issues with improved correspondence strategies, its design is not tailored for persistent SLAM. Later, VGGT-SLAM (Maggio et al., 2025) builds on the strong backbone of VGGT (Wang et al., 2025a), feeding submaps into it and optimizing poses on the $SL(4)$ manifold to achieve more accurate tracking.

**Monocular GS-SLAM.** 3D Gaussian Splatting (3DGS) (Kerbl et al., 2023) has recently gained attention in monocular SLAM research due to its differentiable nature and real-time rendering efficiency. MonoGS (Matsuki et al., 2024) and PhotoSLAM (Huang et al.) are early monocular GS-SLAM methods that initialize Gaussian ellipsoids through feature points or random sampling and incorporate ORB-SLAM3 (Mur-Artal et al., 2015) for pose estimation, enabling applications in small indoor environments. SEGS-SLAM (Tianci Wen, 2025) further enhances structural consistency by modeling appearance variations. DroidSplat (Homeyer et al., 2025) leverages dense optical flow and depth priors to achieve robust tracking and reconstruction. However, these monocular systems suffer from scalability issues and often generate floating artifacts in dynamic or large-scale scenes. Building upon these limitations, approaches like WildGS-SLAM (Zheng et al., 2025), DepthGS (Zhao et al., 2025), and Dy3DGS-SLAM (Li et al., 2025) introduced geometry prior and pixel-level uncertainty estimation to enhance robustness in real-world dynamic scenes. Furthermore, S3PO-GS (Cheng et al., 2025) addresses the challenges of scale drift and the lack of geometric priors commonly encountered in outdoor scenarios by introducing a scale self-consistent pointmap. However, existing GS-based SLAM methods are generally limited to around 1 FPS, which is clearly insufficient to meet the inherent real-time requirements of SLAM. The main reason lies in the fact that these methods train the Gaussians from scratch for each keyframe, typically requiring tens to hundreds of iterations. Since a single iteration takes approximately 20 ms, the total training time per keyframe is roughly one second, inevitably resulting in slow overall performance.

## 3 PRELIMINARIES: 2D GAUSSIAN FOR GEOMETRIC ACCURACY

The original 3D Gaussian Splatting (3DGS) (Kerbl et al., 2023) often produces noisy geometry with "floater" artifacts, as its volumetric primitives lack explicit surface constraints. To address this, 2D Gaussian Splatting (2DGS) was introduced in (Huang et al., 2024), representing scenes as a collection of 2D planar Gaussian surfels. Their work demonstrated that this representation provides stronger geometric priors, yielding significantly improved surface accuracy and multi-view consistency over 3DGS.

We adopt 2DGS as scene representation, where each surfel is defined by its position ($\boldsymbol{\mu}$), color ($\boldsymbol{c}$), opacity ($\sigma$), rotation ($\boldsymbol{r}$), and 2D scale ($\boldsymbol{s}$). The final pixel color ($\hat{I}$), depth ($\hat{D}$), and accumulation

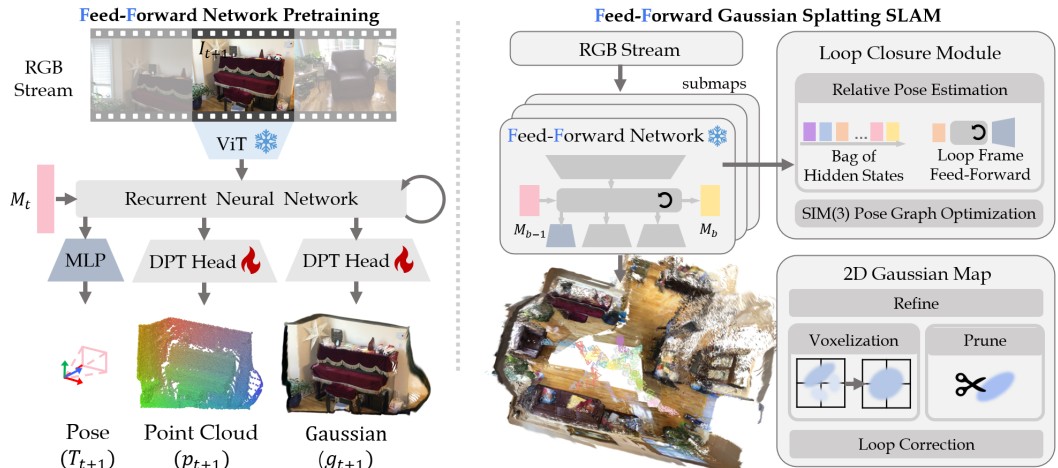

Figure 2: **Pipeline.** For each new frame, our recurrent model jointly infers the camera pose and per-pixel 2DGS attributes conditioned on a hidden state. The hidden state is updated simultaneously. To avoid catastrophic forgetting, the stream is partitioned into submaps. The hidden state is reinitialized for each submap. Past hidden states are cached in the Bag of Hidden States. Upon loop detection, i.e., revisiting a location, we perform a single forward pass on the loop frame conditioned on the past hidden state to relocalize the current frame in the past submap. A following pose graph optimization is then performed to correct the full trajectory. In the backend, per-frame 2DGS attributes prediction is voxelized, merged, and refined to build a global 2DGS map.

$(\hat{A})$ are rendered via volumetric alpha blending:

$$w_i(p) = \sigma_i \cdot \exp\left(-\frac{1}{2}(p - \boldsymbol{\mu}_i)^T \Sigma_i^{-1}(p - \boldsymbol{\mu}_i)\right)$$
$$(\hat{I}, \hat{D}, \hat{A}) = \sum_{i=1}^{N}(\boldsymbol{c}_i, \boldsymbol{z}_i, 1)w_i \prod_{j=1}^{i-1}(1 - w_j) \tag{1}$$

Here, $p$ denotes a pixel coordinate, $\Sigma_i \in \mathbb{R}^{2\times2}$ is the screen-space covariance induced by the surfel's rotation $\boldsymbol{r}_i$ and 2D scale $\boldsymbol{s}_i$, and $\boldsymbol{z}_i$ is the surfel depth along the camera optical axis used for depth accumulation. Compared to 3D Gaussian ellipsoids, the planar 2DGS representation provides a stronger surface prior that suppresses floaters and improves geometric fidelity, which is particularly beneficial for SLAM where small geometric inconsistencies can quickly accumulate into drift. In the remainder of this paper, we use 2DGS as our map primitive: our recurrent feed-forward frontend predicts per-pixel surfel attributes in the current camera frame, and our backend incrementally fuses and refines these predictions into a global, renderable map that can be efficiently updated after pose graph optimization.

## 4 OUR APPROACH

In this section, we introduce our approach in the following order. We first describe our recurrent feed-forward frontend, which constitutes the core of our system by incrementally estimating camera poses and per-frame 2DGS attributes (§4.1). We then present our loop closure mechanism, which leverages the model's hidden state to enable global drift correction via Sim(3) optimization (§4.2). Finally, we detail the backend mapping method that incrementally fuses the frontend's raw predictions into a globally consistent 2DGS map (§4.3).

### 4.1 RECURRENT FEED-FORWARD FRONTEND MODEL

The input of our system is a monocular RGB stream $\{I_t\}$. For each incoming frame $I_t \in \mathbb{R}^{H\times W\times 3}$ at timestep $t$, our feed-forward model, denoted by $f$, takes the current frame and a hidden state $M_{t-1}$

as input. The function of model $f$ is to jointly predict three outputs: (a) the camera pose $\hat{T}_t \in \text{SE}(3)$, representing the transformation from the current camera frame to the coordinate system of the initial frame ($t = 1$); (b) a dense, pixel-aligned 2DGS map $\hat{\mathcal{G}}_t = \{\mathcal{G}_n\}_{n=1}^{H \times W}$, where the attributes of each Gaussian surfel are defined in the local coordinate system of the current camera; and (c) an updated hidden state $M_t$, which carries aggregated information forward to the next timestep (the initial state $M_0$ is initialized to zero). Formally, the per-frame prediction process is expressed as:

$$\hat{T}_t, \hat{\mathcal{G}}_t, M_t = f(I_t, M_{t-1}) \tag{2}$$

**Model Architecture.** Inspired by Wang et al. (2025b) and Wu et al. (2025b), we design a stateful transformer architecture to incrementally reconstruct the scene. Each incoming image is first converted into a set of visual tokens $F_t \in \mathbb{R}^{K \times C}$ by a ViT encoder. The model then employs two interconnected decoders that facilitate bidirectional information exchange between visual tokens $F_t$ and the persistent hidden state $M_{t-1}$ via cross-attention. A learnable pose token $z_t$, concatenated with $F_t$, is processed by the decoders to aggregate geometric cues for pose estimation. This fusion can be expressed as:

$$F_t = \text{Encoder}(I_t) \tag{3}$$

$$F'_t, z'_t, M_t = \text{Decoders}((F_t, z_t), M_{t-1}) \tag{4}$$

Finally, two DPT heads (Ranftl et al., 2021) decode the image tokens $F'_t$ to predict 2DGS attributes: the means and confidences $\{\hat{\boldsymbol{\mu}}_t, \hat{C}_t\}$, and other parameters $\{\hat{\boldsymbol{\sigma}}_t, \hat{\boldsymbol{r}}_t, \hat{\boldsymbol{s}}_t, \hat{\boldsymbol{c}}_t\}$. Concurrently, an MLP head extracts the absolute camera pose $\hat{T}_t$ from the output pose token $z'_t$.

$$\hat{\boldsymbol{\mu}}_t, \hat{C}_t = \text{Head}_{\text{means}}(F'_t) \tag{5}$$

$$\hat{\boldsymbol{\sigma}}_t, \hat{\boldsymbol{r}}_t, \hat{\boldsymbol{s}}_t, \hat{\boldsymbol{c}}_t = \text{Head}_{\text{gs}}(F'_t) \tag{6}$$

$$\hat{T}_t = \text{Head}_{\text{pose}}(z'_t) \tag{7}$$

**Training Objective.** Our model is trained on large-scale datasets with ground-truth RGB, depth, and camera pose data. The training objective consists of three loss components, summed over a sequence of length $L$. The predicted pose $\hat{T}_t$ is parameterized as a quaternion $\hat{q}_t$ and a translation vector $\hat{\tau}_t$. The total loss is a weighted sum of the pose loss, geometric loss, and rendering loss:

$$\mathcal{L}_{\text{total}} = \lambda_{\text{pose}}\mathcal{L}_{\text{pose}} + \lambda_{\text{geo}}\mathcal{L}_{\text{geo}} + \mathcal{L}_{\text{render}} \tag{8}$$

$$\mathcal{L}_{\text{pose}} = \sum_{t=1}^{L} (\|\hat{q}_t - q_t\|_2 + \|\hat{\tau}_t - \tau_t\|_2) \tag{9}$$

$$\mathcal{L}_{\text{geo}} = \sum_{t=1}^{L} \sum_{n=1}^{H \times W} (\hat{c}_{t,n} \cdot \|\hat{\mu}_{t,n} - \mu_{t,n}\|_2 - \alpha \log(\hat{c}_{t,n})) \tag{10}$$

$$\mathcal{L}_{\text{render}} = \sum_{t=1}^{L} \left( \lambda_{mse}\|I_t - \hat{I}_t\|_2^2 + \lambda_{lpips}\mathcal{L}_{lpips}(I_t, \hat{I}_t) + \lambda_{depth}\|D_t - \hat{D}_t\|_2^2 \right) \tag{11}$$

where $\hat{I}_t$ and $\hat{D}_t$ are the RGB and depth rendered from the predicted 2DGS map $\hat{\mathcal{G}}_t$ through standard rasterization as described in §3. Here, $(q_t, \tau_t)$, $I_t$, and $D_t$ denote the ground-truth camera pose, RGB image, and depth map, respectively; $\mu$ is obtained by unprojecting the ground-truth depth map using the camera intrinsics. Our model is trained on datasets including DL3DV and ScanNet++, which cover both indoor and outdoor scenes. Please refer to Appendix D for the detailed training setup, and Appendix C.2 for model acceleration.

**Incremental Tracking with Submaps.** While our model can theoretically process an arbitrarily long sequence, we observe in practice that cumulative drift increases with sequence length, $L$, a result of catastrophic forgetting in recurrent models. To ensure robust tracking, we partition the input stream into shorter subsequences (submaps). For each submap, the hidden state is re-initialized; consequently, all predicted poses $\{\hat{T}_t\}$ are expressed in the coordinate frame of that submap's first frame. A one-frame overlap between consecutive submaps allows us to compute the relative transformation between them, enabling us to chain the local pose estimates into a continuous trajectory. This overlap also provides an explicit inter-submap alignment constraint, which is later incorporated into the pose graph for global optimization.

## 4.2 Loop Closure via Hidden State

Monocular SLAM systems inevitably suffer from accumulated pose and scale drift. To address this, we introduce a novel mechanism to compute a geometric constraint between the current frame and a past frame with a single forward pass when the camera revisits a previously mapped area (a loop closure (Tsintotas et al., 2022)). This enables pose graph optimization to produce a globally consistent trajectory, mitigating the long-standing problem of drift.

**Bag of Hidden States as Long-Term Memory.** As our recurrent model processes frames sequentially, the hidden state, $M$, incrementally aggregates local, multi-frame geometric and visual information. The hidden state becomes a rich, contextual summary of the local scene that the system has just observed. We leverage this by caching the final hidden state $M_a$ for each submap $C_a$ in a bag of hidden states. When the system later revisits an area, it can retrieve a historical hidden state from the bag of hidden states to reload past geometric and visual context.

**Loop Frame Feed-Forward for Relocalization.** The process is triggered when a loop candidate is detected between the current keyframe $I_j$ of submap $C_b$ and a historical keyframe $I_i$ of submap $C_a$ using an appearance-based method (Izquierdo & Civera, 2024). We retrieve the cached hidden state $M_a$ associated with the historical submap $C_a$ containing keyframe $I_i$. Intuitively, conditioning on $M_a$ encourages the model to interpret $I_j$ in the coordinate system of the past submap $C_a$, yielding a direct cross-submap constraint. A single forward pass $f(I_j, M_a)$ on the current frame $I_j$ conditioned on this past context yields two key outputs: (1) the relocalized pose $\mathbf{T}_j^a \in \mathrm{SE}(3)$, and (2) the corresponding point cloud interpretation $\mathcal{P}_j^a = \{\boldsymbol{\mu}_k^a\}$. The relative pose is then computed as $\mathbf{T}_{j \to i} = (\mathbf{T}_j^a)^{-1}\mathbf{T}_i^a$. To resolve scale ambiguity, we compare this historical interpretation $\mathcal{P}_j^a$ against $\mathcal{P}_j^b = \{\boldsymbol{\mu}_k^b\}$, which is the point cloud generated from the standard, incremental tracking of frame $I_j$ (i.e., using the current hidden state). Crucially, since both point clouds are in the camera coordinate frame and originate from the same image $I_j$, they differ only by a scale factor. This allows us to robustly solve for the relative scale $s^*$ via least-squares:

$$s^* = \operatorname*{argmin}_s \sum_k \left\| \boldsymbol{\mu}_k^b - s \cdot \boldsymbol{\mu}_k^a \right\|^2. \tag{12}$$

Finally, the estimated scale and relative pose are combined into the complete $\mathrm{Sim}(3)$ loop closure constraint:

$$\mathbf{H}_{j \to i} = \begin{pmatrix} s^* R_{j \to i} & t_{j \to i} \\ \mathbf{0}^T & 1 \end{pmatrix} \tag{13}$$

where $R_{j \to i}$ and $t_{j \to i}$ are the rotation and translation components of $\mathbf{T}_{j \to i}$.

**Pose Graph Optimization.** The computed $\mathrm{Sim}(3)$ constraint enables global optimization of the entire trajectory via a pose graph. In this graph, nodes represent the keyframe poses $\mathbf{T}_k^W \in \mathrm{Sim}(3)$, and edges represent three types of geometric constraints: ***Sequential Constraint***, a factor linking consecutive frames within a submap, computed as the relative transformation $(\mathbf{T}_k^{-1}\mathbf{T}_{k+1})$; ***Inter-Submap Constraint***, an alignment factor connecting adjacent submaps, which is derived by estimating the relative scale between the two point cloud predictions of their shared frame; and our novel ***Loop Closure*** factors that connect distant, revisited parts of the trajectory. The globally optimal set of poses $\mathcal{T}^{W*}$ is found by minimizing a non-linear least-squares cost function over all constraints:

$$\mathcal{T}^{W*} = \operatorname*{arg\,min}_{\mathcal{T}^W} \sum_{(i,j) \in \mathcal{E}} \left\| \log \left( \mathbf{H}_{j \to i}^{-1} \cdot ((\mathbf{T}_i^W)^{-1} \cdot \mathbf{T}_j^W) \right) \right\|_\Omega^2 \tag{14}$$

where the residual error is computed in the Lie algebra $\mathfrak{sim}(3)$ using the logarithmic map $\log(\cdot)$. This formulation finds the trajectory that best satisfies all geometric constraints simultaneously. We solve this efficiently using GTSAM (Dellaert & Contributors, 2022), and the resulting corrected poses are passed to the backend to update the 2DGS map, as detailed in §4.3.

## 4.3 2DGS Map Optimization

The backend runs in a separate thread and incrementally builds and optimizes a globally consistent 2DGS map. For each new keyframe, it takes as input the RGB image $I_k$, the globally optimized camera pose $T_k \in \mathrm{Sim}(3)$, and the per-pixel 2DGS map $\hat{\mathcal{G}}_k$ of $I_k$ predicted by the frontend. The backend

pipeline then consists of four key stages: (1) pre-processing the dense predictions via adaptive voxelization; (2) merging the 2DGS map of new frame into the global map; (3) applying a lightweight local refinement; and (4) executing global map corrections after a successful loop closure.

**Adaptive Voxelization.** We empirically found that the per-pixel 2DGS predicted by the frontend is sometimes overly dense. To reduce memory consumption, we first process each incoming 2DGS map $\hat{\mathcal{G}}_k$ with an adaptive voxelization filter prior to merging. The map is partitioned into blocks of $2 \times 2$ 2DGS primitives. Primitives within each block are consolidated into a single merged primitive by averaging their attributes:

$$\boldsymbol{\theta}_{\text{merged}} = \frac{1}{N} \sum_{n=1}^{N} \boldsymbol{\theta}_n, \quad \text{for } \boldsymbol{\theta} \in \{\boldsymbol{\mu}, \boldsymbol{\sigma}, \boldsymbol{c}, \boldsymbol{s}\}. \tag{15}$$

$$\boldsymbol{r}_{\text{merged}} = \frac{\sum_{n=1}^{N} \text{align}(\boldsymbol{r}_n, \boldsymbol{r}_1)}{\|\sum_{n=1}^{N} \text{align}(\boldsymbol{r}_n, \boldsymbol{r}_1)\|} \tag{16}$$

where $\text{align}(\cdot)$ is the standard process to ensure consistent quaternion averaging. To preserve geometric details, blocks with a depth variation exceeding a threshold $\tau_d$ are excluded from this process.

**Map Fusion.** With each new frame, we first maintain the existing global map by pruning erroneous Gaussians. This is done by rendering the map from the current camera pose $T_k$ using the formula from §3, and removing any primitive that contributes to pixels with high RGB or depth reconstruction error. Subsequently, the incoming voxelized 2DGS primitives are fused into the global map. First, they are transformed from camera to world coordinates:

$$\mu_{\text{world}} = s_k R_k \mu_{\text{cam}} + t_k, \quad r_{\text{world}} = q_k \cdot r_{\text{cam}} \tag{17}$$

where $(s_k, R_k, t_k)$ are components of $T_k$; $q_k$ is the quaternion form of $R_k$. To avoid redundant densification, we only add these new primitives in regions that are not yet well-reconstructed, identified by rendering an accumulation map from the global map and checking against a threshold $\tau_{accum}$.

**Lightweight Map Refinement.** A key advantage of our *Predict-and-Refine* paradigm is the dramatically reduced optimization workload for the backend. The high-quality per-frame predictions from our frontend serve as a strong geometric and appearance prior. Consequently, after the fusion step, we only need to refine the local map region associated with the latest $K$ keyframes for only 20 iterations. This stands in stark contrast to existing 3DGS-SLAM methods that require hundreds to thousands of optimization iterations per frame, especially during initialization.

**Loop Correction of Gaussian Map.** As described in §4.2, upon receiving the set of globally optimized poses $\mathcal{T}^{W*}$ after a loop closure, the backend initiates map correction to ensure the 2DGS map aligns with the corrected trajectory. A naive approach would be to re-run rendering-based optimization using the corrected poses, but we found this to be prohibitively slow. Therefore, we adopt a more efficient rigid transformation strategy. We rigidly bind the 2DGS primitives to their originating keyframe. When a keyframe's pose is updated from $T_{\text{old}}$ to $T_{\text{new}}$, we compute the delta transformation $\Delta T = T_{\text{new}} \cdot T_{\text{old}}^{-1}$ and apply it to all associated primitives. This process efficiently warps the map to align with the corrected trajectory without costly re-rendering.

## 5 EXPERIMENTS

### 5.1 EXPERIMENTAL SETUP

We evaluate our system on three challenging real-world datasets: ScanNet (Dai et al., 2017a), BundleFusion (Dai et al., 2017b), and KITTI (Geiger et al., 2012). ScanNet and BundleFusion consist of large-scale indoor scenes with motion blur and diverse lighting conditions. We treat ScanNet as in-domain evaluation and BundleFusion as out-of-domain evaluation. KITTI features large-scale outdoor driving scenarios with high scale variance and dynamic objects. We evaluate tracking accuracy using Absolute Trajectory Error (ATE RMSE) and rendering quality via PSNR, SSIM, and LPIPS. Since monocular SLAM has inherent scale ambiguity, we compute ATE after $\text{Sim}(3)$ alignment to ground truth. For ScanNet and BundleFusion, we further evaluate geometric quality with scale-aligned Depth $L1$ error. All experiments are conducted on a single RTX 4090 GPU paired with an Intel Xeon 6133 CPU (2.50GHz).

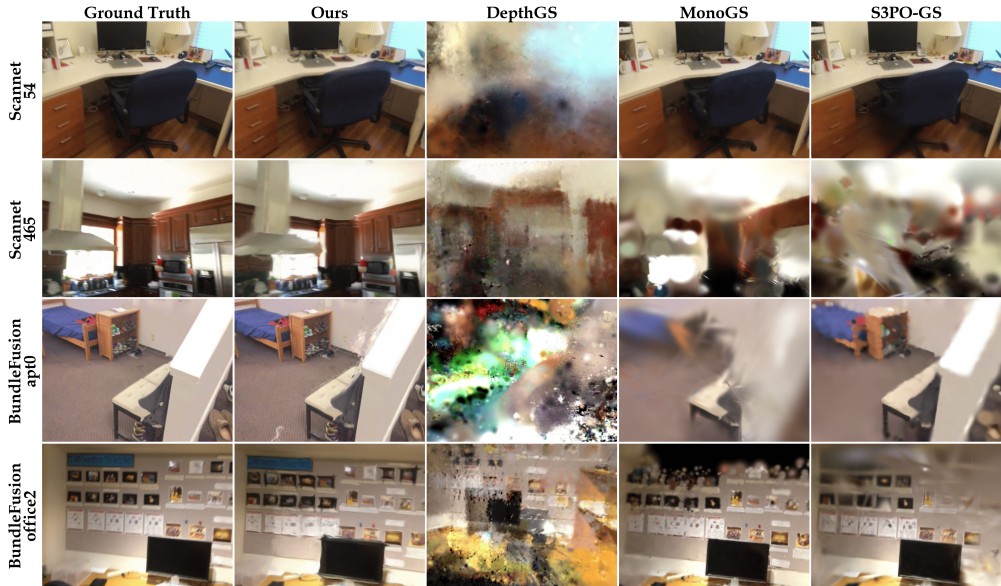

Figure 3: **Qualitative Rendering Results.**

**Baselines.** We compare Flash-Mono with three state-of-the-art monocular GS-SLAM systems on both mapping and tracking quality: MonoGS (Matsuki et al., 2024), DepthGS (Zhao et al., 2025), and S3POGS (Cheng et al., 2025). We also compare against leading monocular SLAM systems renowned for pose accuracy, although they do not produce dense renderings, including ORB-SLAM3 (Campos et al., 2021), DROID-SLAM (Teed & Deng, 2021), and MASt3R-SLAM (Murai et al., 2025). On KITTI, we primarily compare against S3POGS, as we encountered frequent failures while evaluating other indoor-focused GS-SLAM baselines due to the large-scale and high dynamic nature of KITTI.

## 5.2 TRACKING PERFORMANCE

As shown in Table 1, Flash-Mono significantly outperformed all traditional and GS-SLAM baseline methods. On most scenes, we also surpassed MASt3R-SLAM, a recent feed-forward SLAM system. This validates the effectiveness of multi-frame context and our novel hidden-state-based relocalization mechanism.

Table 1: ATE RMSE (cm) on **ScanNetV1** and **BundleFusion** datasets. Lower is better. We mark the **first** and second best results.

| ATE [cm]↓ | ScanNetV1 | | | | | | BundleFusion | | | | |
|---|---|---|---|---|---|---|---|---|---|---|---|
| | **0054** | **0059** | **0106** | **0169** | **0233** | **0465** | **apt0** | **apt2** | **copyroom** | **office0** | **office2** |
| ORB-SLAM3 | 243.26 | 90.67 | 178.13 | 60.15 | 25.01 | 181.86 | 87.37 | 265.64 | 27.60 | 116.33 | 49.33 |
| DROID-SLAM | 161.22 | 69.92 | 89.11 | 28.26 | 74.01 | 117.27 | 89.38 | 148.04 | 19.71 | 31.41 | 73.91 |
| MonoGS | 70.19 | 97.24 | 150.89 | 191.98 | 62.45 | 113.19 | 122.59 | 142.54 | 53.41 | 62.67 | 127.02 |
| DepthGS | 192.18 | 93.69 | 140.19 | 205.92 | 81.90 | 121.01 | 67.52 | 119.74 | 14.59 | 40.42 | 16.05 |
| S3PO-GS | 69.36 | 16.52 | 26.15 | 87.04 | 27.09 | 96.35 | 92.49 | 97.90 | 21.88 | 64.22 | 69.88 |
| MASt3R-SLAM | 13.25 | 10.89 | 15.83 | 15.24 | **10.99** | 15.74 | **9.65** | 13.66 | 9.28 | 9.97 | 9.92 |
| Ours | **11.69** | **8.89** | **10.83** | **10.16** | 12.13 | **13.00** | 11.44 | **12.36** | **7.34** | **8.74** | **9.34** |

## 5.3 MAPPING PERFORMANCE

Table 2 presents the rendering quality results. Although we perform only 20 optimization iterations per keyframe (a **10x** reduction compared to the 250 iterations used by MonoGS (Matsuki et al., 2024) and S3PO-GS (Cheng et al., 2025)), our method achieves superior or competitive rendering quality.

Table 2: Mapping quality on **ScanNetV1** and **BundleFusion**. Higher is better for SSIM/PSNR, lower is better for LPIPS. We mark the **first** and second best results.

| Method | Metric | ScanNetV1 | | | | | | | BundleFusion | | | | | |
|---|---|---|---|---|---|---|---|---|---|---|---|---|---|---|
| | | 0054 | 0059 | 0106 | 0169 | 0233 | 0465 | FPS↑ | apt0 | apt2 | copyroom | office0 | office2 | FPS↑ |
| **MonoGS** | SSIM↑ | **0.80** | **0.74** | 0.72 | 0.77 | 0.68 | 0.59 | | 0.70 | 0.39 | 0.70 | 0.52 | 0.60 | |
| | LPIPS↓ | 0.61 | 0.60 | 0.54 | 0.66 | 0.67 | 0.74 | 0.69 | 0.67 | 0.82 | 0.63 | 0.78 | 0.71 | 1.00 |
| | PSNR↑ | 19.24 | 16.54 | 16.09 | **18.86** | 17.65 | 14.52 | | 13.68 | 11.50 | 14.37 | 13.38 | 13.96 | |
| **DepthGS** | SSIM↑ | 0.31 | 0.32 | 0.34 | 0.42 | 0.36 | 0.26 | | 0.38 | 0.41 | 0.58 | 0.56 | 0.58 | |
| | LPIPS↓ | 0.79 | 0.78 | 0.78 | 0.73 | 0.84 | 0.81 | 1.57 | 0.67 | 0.69 | 0.51 | 0.62 | 0.63 | 1.28 |
| | PSNR↑ | 12.29 | 12.42 | 11.76 | 13.64 | 13.17 | 11.11 | | 13.65 | 14.85 | 17.00 | 15.96 | 16.51 | |
| **S3PO-GS** | SSIM↑ | **0.80** | 0.71 | **0.75** | **0.78** | **0.73** | 0.61 | | **0.74** | **0.64** | 0.47 | 0.63 | **0.64** | |
| | LPIPS↓ | 0.62 | 0.58 | 0.54 | 0.55 | 0.69 | 0.75 | 0.71 | 0.57 | 0.71 | 0.78 | 0.71 | 0.64 | 0.94 |
| | PSNR↑ | 20.79 | 17.19 | 17.60 | 18.52 | 18.37 | 14.14 | | 18.98 | 15.72 | 18.56 | 15.23 | 16.59 | |
| **Ours** | SSIM↑ | 0.79 | 0.66 | **0.72** | 0.73 | 0.69 | **0.66** | | 0.66 | 0.60 | 0.72 | **0.69** | **0.64** | |
| | LPIPS↓ | **0.39** | **0.41** | **0.43** | **0.39** | **0.44** | **0.45** | 12.71 | **0.49** | **0.54** | **0.45** | **0.50** | **0.51** | 11.99 |
| | PSNR↑ | **21.73** | **17.83** | **17.75** | 18.52 | **21.60** | **19.51** | | **19.03** | **16.48** | **19.50** | **17.10** | **17.63** | |

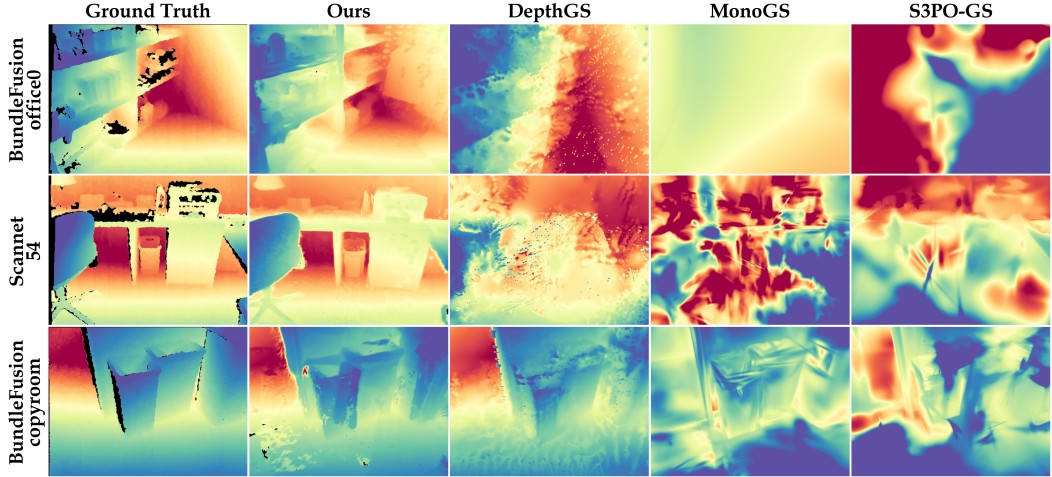

Figure 4: **Qualitative Analysis on Rendered Depth.**

This highlights the effectiveness of our *Predict-and-Refine* paradigm: high-quality Gaussians predicted by our foundation model reduce the need for costly backend optimization. The scale-aligned Depth $L1$ error is evaluated in Table 5. We achieve a lower Depth L1 error, suggesting a more accurate underlying 3D scene reconstruction. Qualitative rendered RGB and depth are presented in Figure 3 and Figure 4.

## 5.4 OUTDOOR EVALUATION ON KITTI

We further evaluate Flash-Mono on the KITTI benchmark to assess generalization to large-scale outdoor environments. Since MonoGS and DepthGS are designed primarily for indoor scenes, they often fail under the large scale variance and dynamics in KITTI; therefore, we mainly compare with S3PO-GS (Cheng et al., 2025), which is designed for outdoor scenarios. Table 3 reports tracking accuracy and Table 4 reports rendering quality.

Table 3: ATE RMSE (m) on **KITTI Odometry**. Lower is better.

| ATE RMSE [m]↓ | 00 | 05 | 06 | 07 | 08 | 28 |
|---|---|---|---|---|---|---|
| **Ours** | **12.85** | **16.58** | **9.93** | **12.08** | **45.25** | **16.75** |
| **S3PO-GS** | 32.49 | 34.76 | 16.43 | fail | 64.74 | 23.64 |

Table 4: Rendering quality on **KITTI Odometry**. Higher is better for PSNR/SSIM, lower is better for LPIPS.

| Method | Metric | 00 | 05 | 06 | 07 | 08 | 28 |
|---|---|---|---|---|---|---|---|
| **S3PO-GS** | PSNR ↑ | 16.65 | 15.64 | 13.55 | fail | **17.25** | 15.30 |
| | SSIM ↑ | 0.5409 | 0.5320 | 0.4726 | fail | 0.5912 | 0.5053 |
| | LPIPS ↓ | 0.6254 | 0.6352 | 0.7241 | fail | **0.4626** | 0.6131 |
| **Ours** | PSNR ↑ | **17.41** | **17.01** | **15.13** | **17.89** | 16.12 | **17.47** |
| | SSIM ↑ | **0.6584** | **0.6278** | **0.5922** | **0.6036** | **0.6221** | **0.5633** |
| | LPIPS ↓ | **0.5358** | **0.4871** | **0.5333** | **0.4854** | 0.4710 | **0.4581** |

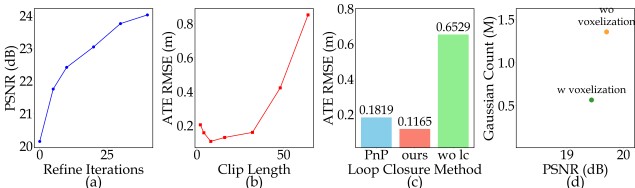

Figure 5: **Ablation studies.** (a) Refine Iterations vs. PSNR. (b) Submap Length vs. ATE RMSE. (c) Loop Closure Settings. (d) PSNR vs. Model Size.

Table 5: Mean Depth L1 Error (m) on **ScanNet** and **BundleFusion**. We mark the **best** results.

| L1(m) ↓ | Scan. | Bundle. |
|---|---|---|
| MonoGS | 1.19 | 1.20 |
| DepthGS | 0.49 | 0.23 |
| S3PO-GS | 0.52 | 0.85 |
| Ours | **0.34** | **0.21** |

## 5.5 ABLATION

We conducted ablation studies to analyze the impact of key system components. The results are shown in Figure 5. First, we evaluated the effect of backend refinement iterations on rendering quality (PSNR). Without refinement (0 iterations), the direct output from our feed-forward model achieves a PSNR of 20.14. Applying 10 refinement iterations increases the PSNR to 22.41, indicating that the model provides a strong initial prediction that can be efficiently improved by a few optimization steps. Second, we examined the influence of submap clip length on tracking accuracy (ATE RMSE). The lowest error of 0.106 was observed with a clip length of 8 frames. Shorter lengths resulted in higher error, suggesting insufficient temporal context, while lengths greater than 16 frames also increased the error, which points to the accumulation of intra-submap drift caused by the forgetting characteristic of RNN models. This supports the strategy of partitioning the input stream. Third, we compared our hidden state-based loop closure against a traditional PnP+RANSAC baseline and a configuration with no loop closure. Our original system beats the other two settings on tracking performance by a large margin, suggesting our approach generates more accurate $Sim(3)$ constraints. Finally, our adaptive voxelization module reduced the total number of Gaussian primitives by over 58% (from 1.35M to 0.56M), which corresponded to a minor PSNR decrease from 19.70 to 19.44. This demonstrates the module's role in creating a more compact map representation at a small cost to rendering fidelity.

## 6 CONCLUSION

We presented Flash-Mono, a real-time monocular Gaussian Splatting SLAM system that fundamentally shifts from the time-consuming *Train-from-Scratch* paradigm to an efficient *Predict-and-Refine* approach. As our experiments show, Flash-Mono achieves state-of-the-art rendering quality with a **10x** reduction in computation time. Furthermore, we introduced a novel loop closure mechanism that enables robust $Sim(3)$ optimization to correct scale and pose drift inherent in monocular systems, leading to superior tracking accuracy on complex indoor scenes.

## ACKNOWLEDGMENTS

This work was supported in part by the National Natural Science Foundation of China (NSFC) under Grant 62403142, and in part by the Science and Technology Commission of Shanghai Municipality under Grant 24511103100.

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

## A  LLM USAGE STATEMENT

In the preparation of this manuscript, we utilized Large Language Models (LLMs) as assistive tools, in accordance with the ICLR policy. The specific roles of these models are detailed below.

We employed **GPT-5** primarily for **language polishing and refinement**. After drafting the paper, we used the model to improve grammar, clarity, and overall readability. The core ideas, experimental setup, results, and conclusions were conceived and articulated entirely by the human authors. We reviewed and edited all model-generated suggestions to ensure the final text accurately reflects our original research and contributions.

We also used **Gemini 2.5 Pro** with its deep research capabilities to assist in the **literature review process**. This tool helped in identifying relevant prior work, summarizing existing literature, and locating publicly available publications. All cited works were subsequently read, analyzed, and contextualized by the authors to build the foundation for our research.

The authors take full responsibility for all content presented in this paper, including its scientific validity, accuracy, and originality. LLMs were used strictly as productivity tools and are not considered authors of this work.

## B  MORE EXPERIMENTAL SETUP AND RESULTS

### B.1  BASELINE

For MASt3R-SLAM (Murai et al., 2025), we adopted the official experimental configuration with $\omega_k = 0.333, \omega_l = 0.1, \omega_r = 0.005$, and a maximum of 10 matching iterations. For MonoGS (Matsuki et al., 2024), we followed their TUM settings, as TUM shares the closest characteristics with the BundleFusion (Dai et al., 2017b) and ScanNetV1 (Dai et al., 2017a) datasets. During evaluation, MonoGS encountered out-of-memory (OOM) failures on three sequences: ScanNet 0054 and 0465, and the apt0 sequence in BundleFusion. For these cases, the reported metrics are computed only on the subsequences successfully reconstructed before the crash. This truncation may lead to an optimistic bias, as drift accumulation over the full sequence would likely degrade reconstruction and rendering quality further. For S3PO-GS (Cheng et al., 2025), we used their official base configuration while loading the ground-truth intrinsics for the test datasets. For the ScanNetV1 sequence 0465, accumulated pose drift caused $PnP$ failure, and results are reported only on the valid subsequence. For DepthGS (Zhao et al., 2025), we followed the official repository guidelines, generating monocular depth maps for each sequence using the *UniDepthV2-large* checkpoint and benchmarking under the provided experimental settings. Importantly, the reported FPS includes the runtime required for UniDepthV2 (Piccinelli et al., 2025), ensuring a fair comparison across methods.

### B.2  COMPARISON DETAILS ON DEPTH RENDERING QUALITY

As shown in Table 6, we record the depth rendering results in detail. To avoid bias caused by large errors in failure scenarios, we report in the main text the mean values excluding the maximum and minimum.

Table 6: Depth L1 Error (m) on **ScanNetV1** and **BundleFusion** datasets. Lower is better. We mark the best results.

| Depth L1 [m] ↓ | ScanNetV1 | | | | | | BundleFusion | | | | |
|---|---|---|---|---|---|---|---|---|---|---|---|
| | 0054 | 0059 | 0106 | 0169 | 0233 | 0465 | apt0 | apt2 | copyroom | office0 | office2 |
| **MonoGS** | 1.06 | 1.27 | 1.41 | 1.56 | 0.89 | 0.82 | 0.96 | 1.28 | 1.18 | 1.15 | 1.26 |
| **DepthGS** | 0.45 | 0.66 | 0.52 | 0.48 | 0.41 | 0.47 | 0.37 | **0.29** | 0.13 | 0.18 | 0.21 |
| **S3PO-GS** | 0.58 | 0.35 | 0.55 | 0.66 | 0.28 | 0.89 | 0.72 | 0.99 | 0.41 | 1.01 | 0.85 |
| **Ours** | **0.16** | **0.23** | **0.51** | **0.17** | **0.35** | **0.44** | **0.33** | 0.35 | **0.11** | **0.11** | **0.18** |

### B.3 MORE QUALITATIVE RESULTS

Figure 6 provides a qualitative comparison of camera trajectories from different methods. We plot the estimated trajectory (colored line) against the ground truth (dashed gray line), projected onto the XY plane. The color of the path indicates the magnitude of the error, following a gradient from blue (low error) to red (high error).

Figure 7 provides a qualitative comparison of the reconstructed map on ScanNet scene 0054. Scene 0054 is a multi-room apartment with varying lighting conditions. All baselines failed to reconstruct the scene.

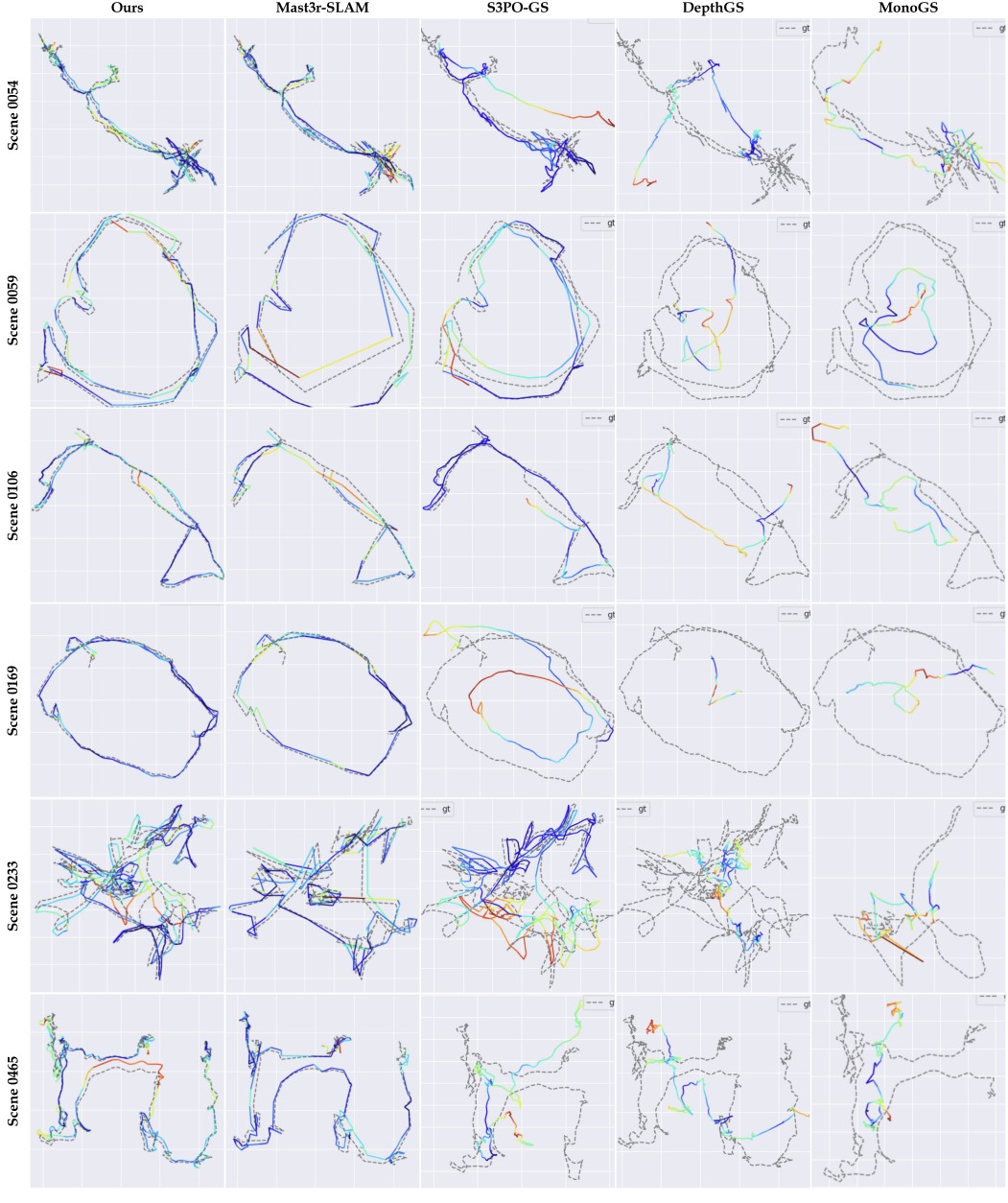

Figure 6: **Qualitative Analysis on Estimated Trajectory**

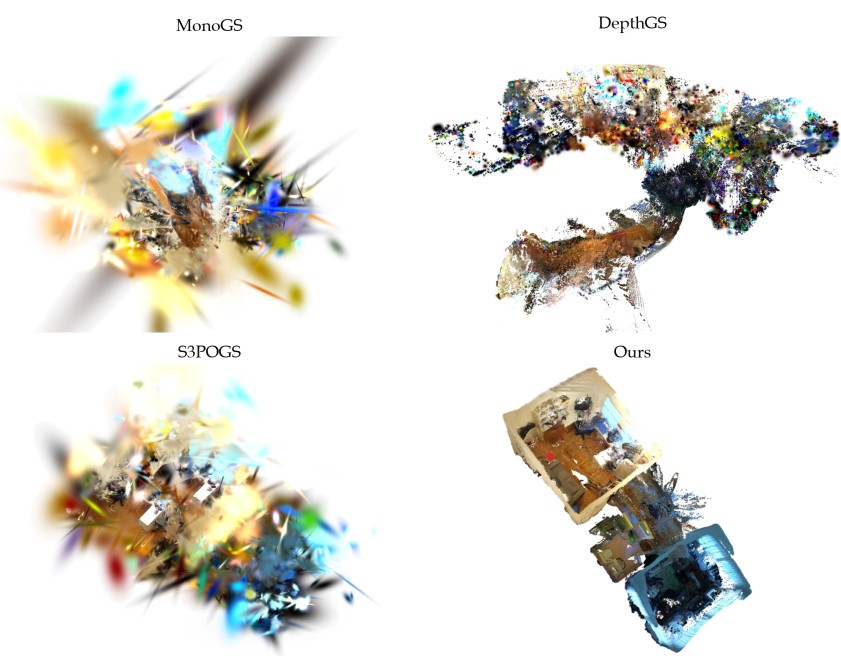

Figure 7: **Qualitative Analysis on reconstructed ScanNet scene 0054.** All baselines failed to reconstruct the scene.

## C   MODEL SIZE AND ACCELERATION

### C.1   MODEL SIZE

To address concerns regarding the feasibility of deployment on resource-constrained devices (e.g., edge devices or laptops), we provide a detailed breakdown of our model size and a performance analysis on lower-end hardware.

**Model Size and Memory Usage.** The detailed parameter breakdown of our architecture is presented in Table 7. The complete model consists of 795.7 million parameters. In terms of memory consumption, the system requires approximately **3GB of VRAM** to run during inference.

Table 7: Detailed breakdown of Flash-Mono model parameters.

| Component | Total Parameters |
|---|---|
| Encoder | 303.1 M |
| Decoder | 380.8 M |
| Heads & Tokens | 111.8 M |
| **Total** | **795.7 M** |

### C.2   MODEL ACCELERATION

The feed-forward frontend is the primary computational bottleneck in the Flash-Mono system. To enhance its practicality for SLAM applications on more accessible, resource-constrained hardware, we tested the effectiveness of several acceleration methods.

First, we converted the attention module parameters from float32 to float16 precision. This strategy compresses the model size and accelerates inference without degrading downstream task accuracy. Second, we addressed an inefficiency in the single-image inference pipeline. With a batch size of 1,

frequent CPU-side operator launches create a bottleneck that underutilizes the GPU. By employing CUDA Graphs, we merged multiple operator calls into a single, efficient launch. See Figure 8.

To validate these improvements under a resource-constrained SLAM setting, we benchmarked the system on a laptop version NVIDIA RTX 4060 GPU (8GB). These optimizations reduced frontend inference latency from 283 ms to just 85 ms, a **3.33× speedup**. Notably, the inference time on the laptop RTX 4060 after acceleration (83 ms) is comparable to the inference time on the high-end RTX 4090 (24GB) used in our main experimental setup (62 ms). In addition, as our model is based on the transformer architecture, further optimizations, such as quantization and efficient attention mechanisms, remain promising directions for future inference acceleration.

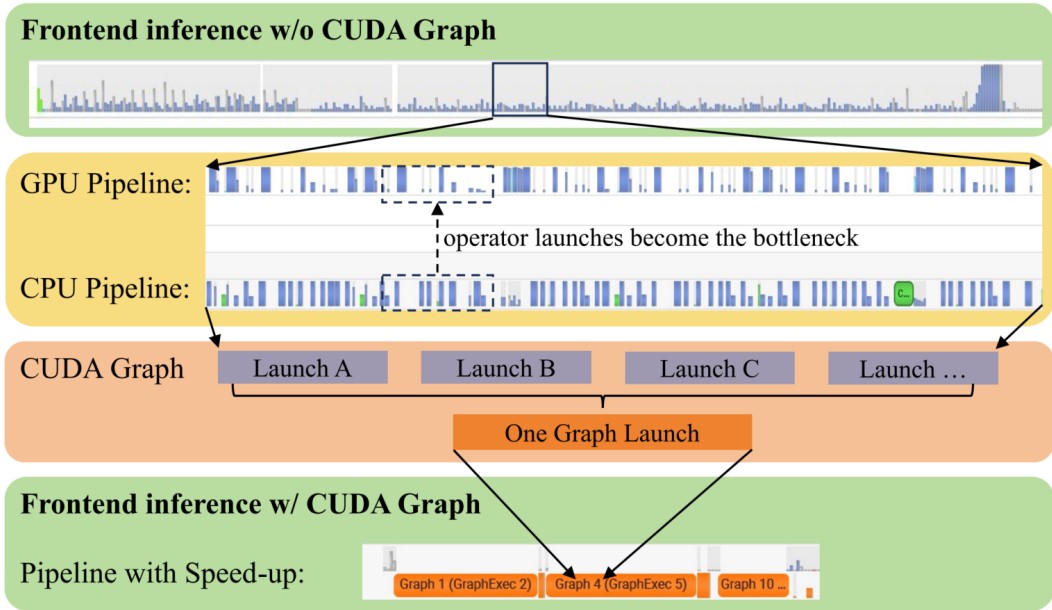

Figure 8: CUDA Graph optimization

# D    TRAINING SETUP

## D.1    DATASETS

We train our model on a combination of indoor and outdoor datasets, including ScanNet++, DL3DV, and Replica. For each training sequence, we utilize the provided RGB video stream, ground truth camera poses, and depth maps. The ground truth point cloud($\mu_t$) required for supervising the geometry loss is generated by unprojecting the ground truth depth map $D_t$ using the corresponding camera intrinsics $K$.

## D.2    EXTRA RENDERING LOSS

While the pose and geometry loss terms adhere to the standard formulations outlined in the main paper, the rendering loss incorporates a more sophisticated strategy. Our empirical investigation revealed that a naive rendering loss, computed solely on the merged Gaussian point cloud from an entire sequence, encourages the model to excessively shrink the scale of individual Gaussians to prevent inter-frame conflicts. In our incremental SLAM setting, this will lead to a bad rendering result on the first few frames of each submap. Thus, our rendering loss consists of two parts with the same weights. The first is a per-frame rendering loss, where the predicted 2DGS map for each frame, $\mathcal{G}_t$, is rendered independently and compared against its corresponding ground truth image and depth. For the second rendering loss, the 2DGS predictions from the entire input sequence, $\{\mathcal{G}_t\}_{t=1}^N$, are merged into a global 2DGS map and then rendered and supervised against the ground truth RGB and depth.

### D.3 Training Curriculum

We designed a three-stage training curriculum. We first warm up the GS head for 5,000 steps. In this initial phase, we initialize our model parameters from CUT3R (Wang et al., 2025b) and freeze all network parameters except for the final 2DGS attribute prediction head and employ a relatively high learning rate of $2 \times 10^{-4}$ on short input sequences of 1-4 frames. The freezing prevents the gradients from excessive rendering loss from backpropagating into the well-trained model backbone. Following this, we unfreeze the parameters of the decoder, the pose head, and the point-mean prediction head, while significantly reducing the learning rate to $1 \times 10^{-5}$ and fixing the sequence length at 4 frames. This intermediate stage is intended to allow the model to adapt to the specific data distribution of the task and mitigate the risk of gradient explosion common in recurrent architectures. Finally, we adapt the model to a longer sequence. The learning rate is further decreased to $5 \times 10^{-6}$, and the maximum sequence length is extended to 32 frames.

## E Detailed Runtime Breakdown

To substantiate the claimed efficiency and clarify the FPS calculation protocol mentioned in the main paper, we provide a comprehensive runtime breakdown of Flash-Mono.

**FPS Calculation Protocol.** The reported FPS is an end-to-end metric, calculated as $\frac{\text{Total Frames}}{\text{Total Runtime}}$. This metric explicitly accounts for **all** system components, including frontend inference, backend map refinement, loop closure detection, pose graph optimization (PGO), and other system overheads.

**Runtime Analysis.** Flash-Mono operates using two parallel threads: a **Frontend** thread responsible for tracking and loop closure, and a **Backend** thread handling mapping and refinement. The detailed time consumption for each module is presented in Table 8.

As shown in the table, the Backend thread (77.5 ms per frame) is slightly slower than the Frontend thread (65 ms per frame), making it the bottleneck of the system. It is important to note that computationally intensive loop closure operations (such as Loop Frame Feedforward and PGO) are sparse events. Consequently, their amortized cost is minimal, allowing the system to maintain high real-time performance.

Table 8: Runtime breakdown of Flash-Mono. The system runs in parallel threads, with the Backend being the primary bottleneck. Loop closure operations are sparse events.

| Thread | Module | Time (ms) | Note |
|---|---|---|---|
| **Frontend** | Feedforward Inference | 62 | Per frame |
| | Loop Closure Detection | 3 | Per frame |
| | **Total (Per Frame)** | **65** | |
| **Backend** | Loop Frame Feedforward | 62 | Per loop closure |
| | PGO (Sim3 Optimization) | 32 | Per loop closure |
| | Merge & Voxelization | 0.5 | Per frame |
| | Refine | 77 | Per frame |
| | **Total (Per Frame)** | **77.5** | |
| | GS Correction | 2 | Per loop closure |

## F Analysis of Gaussian Map Compactness

To evaluate the spatial efficiency of our map representation, we conducted a quantitative analysis of the total number of Gaussian primitives required to represent a complete scene. This metric provides insight into the trade-off between reconstruction quality and memory usage.

Table 9 presents a comparison of the total Gaussian count against several state-of-the-art dense SLAM and Gaussian Splatting methods on three sequences from the TUM RGB-D dataset. As

illustrated in Table 9, Flash-Mono maintains a moderate level of map compactness. The baseline statistics are sourced from CaRtGS (Feng et al., 2024).

Table 9: Quantitative comparison of the total Gaussian count on the TUM dataset. Our method maintains a balance between map density and compactness.

| Method | fr1/desk | fr2/xyz | fr3/office |
|---|---|---|---|
| MonoGS | 26.64k | 43.59k | 35.24k |
| Photo-SLAM | 40.00k | 0.10m | 81.16k |
| SplaTAM | 0.96m | 6.36m | 0.79m |
| Gaussian-SLAM | 0.76m | 0.69m | 1.47m |
| GS-ICP-SLAM | 0.53m | 1.91m | 2.09m |
| **Ours** | **0.63m** | **0.98m** | **0.61m** |

# G ANALYSIS OF HIDDEN STATE FOR LIFE-LONG MAPPING

While the hidden state mechanism has been demonstrated to be highly effective for loop closure relocalization and in-session long-term consistency (as shown in our ablation study in the main paper), we believe this mechanism can be naturally extended to address the challenges of **life-long mapping**—the ability to construct and maintain an up-to-date map as the environment changes over time (e.g., furniture rearrangement, lighting variations, seasonal changes).

**Core Challenges.** Life-long mapping presents two fundamental challenges that must be addressed to maintain a consistent and accurate representation of a dynamic environment:

*Challenge 1: Relocalizing Against an Outdated Map.* The system must be capable of relocalizing a new observation of the changed environment against an old, potentially outdated map. Our hidden state mechanism can address this challenge using the same feed-forward relocalization approach described in Section 4.3 of the main paper.

To demonstrate this capability, we conducted a case study on a scene that underwent significant environmental changes. We first input 8 frames captured at night (with curtains closed and a seat back in place) to generate a hidden state $M_{night}$ representing the historical environment. Subsequently, we fed the model with a new observation of the same scene captured during daytime, where the curtains were open and a person was sitting in the chair. As illustrated in Figure 9, our feed-forward model successfully relocalized the new frame against the outdated hidden state and predicted geometrically consistent results, despite the substantial appearance and content changes.

This result suggests that our current architecture already possesses inherent robustness to environmental variations. We anticipate that training specifically on datasets featuring temporal changes (e.g., time-of-day variations, seasonal shifts) would further enhance this capability.

*Challenge 2: Updating the Map Representation.* Beyond relocalization, the system must update its map representation with new observations to remain synchronized with the current environment state. Our system maintains three core representations: the Gaussian Map, the Pose Graph, and the **Hidden State** (which serves as a compact submap descriptor). While strategies for updating Gaussian primitives (Fu et al., 2025) and pose graphs (Zhao et al., 2021) to changing environment are well-established in the SLAM literature, updating the hidden state descriptor in a life-long setting presents unique challenges.

A naive approach of continuously aggregating new observations into a fixed-capacity hidden state vector inevitably leads to saturation and catastrophic forgetting of historical information. To address this challenge for life-long mapping, we propose two potential strategies:

**1. Discrete State Replacement.** A straightforward yet effective strategy is to detect significant environmental changes during relocalization (e.g., via monitoring the photometric residual or geometric consistency). When substantial changes are identified, rather than attempting to update the obsolete hidden state $M_{old}$, we generate a *fresh* hidden state $M_{new}$ from the current observations. This new state can either replace the outdated state in the Bag of Hidden States or be appended as

an alternative descriptor for the same physical location, effectively maintaining multiple hypotheses (e.g., "daytime" vs. "nighttime" appearance).

**2. Model Adaptation** Instead of discrete replacement, we can implement a continuous update strategy inspired by TTT3R (Chen et al., 2025). This approach reframes the hidden state update as an online learning problem, treating the state as "fast weights" optimized via gradient descent during inference. Crucially, TTT3R introduces a confidence-guided update rule, where the learning rate is dynamically derived from the alignment between the current memory state and the incoming observation. This acts as a self-supervised gating mechanism: it allows the hidden state to selectively integrate persistent environmental changes (where alignment confidence is high) while suppressing transient noise or inconsistent updates (where confidence is low). By adopting this formulation, our system can evolve to capture gradual domain shifts—such as seasonal changes or furniture rearrangement—while mitigating the catastrophic forgetting typically associated with recurrent updates, ensuring the map remains both up-to-date and geometrically consistent over the long term.

**Future Directions.** While our current system demonstrates promising initial capabilities for handling environmental changes, a comprehensive life-long mapping system would require dedicated training on temporally-varying datasets and careful engineering of the state update mechanisms. We believe this represents an exciting direction for future work, building upon the flexible hidden state architecture introduced in this paper.

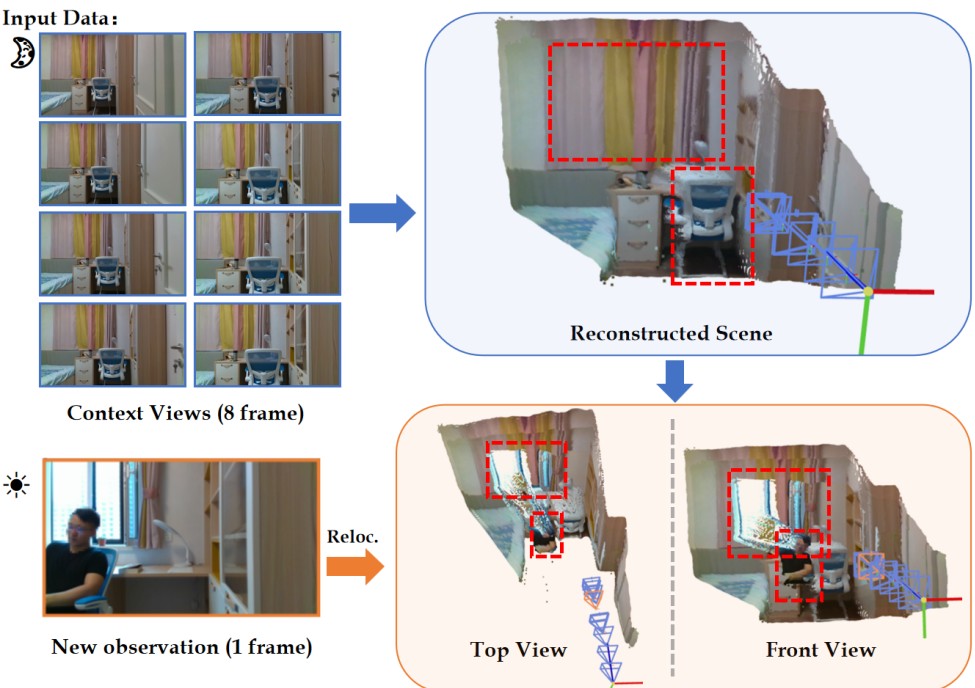

Figure 9: **Case Study: Robust Relocalization Under Environmental Changes.** The model generates a hidden state from 8 context views captured at night (curtains closed, empty chair). When presented with a new observation from the same location but under drastically different conditions (daytime, curtains open, person sitting), the feed-forward model successfully relocalizes and reconstructs accurate geometry. This demonstrates the hidden state's potential for life-long mapping scenarios where environments undergo temporal changes.

