# OpenReview forum: "Flash-Mono: Feed-Forward Accelerated Gaussian Splatting Monocular SLAM"
_ICLR.cc/2026/Conference — ICLR 2026 Poster_

### Official Review · Reviewer_91zN · 2025-10-28

**Soundness:** 3
**Presentation:** 3
**Contribution:** 2
**Rating:** 4
**Confidence:** 4

**Summary:**

The paper presents Flash-Mono, a feed-forward monocular SLAM system that integrates a Transformer-based recurrent frontend, 2D Gaussian Splatting for mapping, and a hidden-state-driven Sim(3) loop closure. It achieves high-quality reconstructions and real-time performance, claiming up to a 10× speedup over optimization-based GS-SLAM methods. Overall, the paper demonstrates strong engineering execution but requires clearer validation to fully substantiate its claims of novelty and robustness.

**Strengths:**

- The paper is generally well written, logically structured, and easy to follow. Figures, tables, and ablations are clearly labeled and help readers understand both the system design and experimental results.

- The paper effectively combines a recurrent feed-forward frontend with a lightweight refinement backend, offering a plausible path toward faster monocular Gaussian Splatting without fully sacrificing reconstruction quality.

- The authors compare against multiple strong baselines (MonoGS, DepthGS, MASt3R-SLAM) using consistent metrics across ScanNet and BundleFusion

- The work reports concrete latency optimizations such as mixed-precision and CUDA Graph execution, showing practical awareness of real-time deployment concerns.

**Weaknesses:**

- Although the paper emphasizes real-time performance and a 10× speedup, the reported FPS is not clearly defined. While Appendix B.1 states that DepthGS includes the UniDepthV2 inference time, it remains unclear whether Flash-Mono’s FPS also accounts for the 20-iteration refinement stage, Sim(3) optimization, loop closure, and rendering overheads. Without these components, the comparison may not reflect true end-to-end latency. A complete runtime breakdown (frontend, refinement, loop closure, rendering) and a unified timing protocol across baselines are necessary to substantiate the claimed efficiency.

- The proposed system integrates several elements already established in previous works: 2DGS, Predict-and-Refine optimization (from existing feed-forward mapping schemes), and Transformer-based hidden state modeling (as seen in MASt3R-SLAM and CUT3R). However, the paper does not sufficiently disentangle which components are novel and which are adapted. The claimed contributions, feed-forward monocular reconstruction and hidden-state-based Sim(3) loop closure, lack rigorous ablation or replacement studies. For instance, the paper does not test substituting 2DGS with 3DGS, removing the hidden state, or comparing Predict-and-Refine with standard local BA. Consequently, the boundary of novelty remains unclear and the contribution feels primarily engineering efforts rather than new approaches.

- The evaluation focuses solely on indoor datasets, i.e. ScanNet V1 and BundleFusion, which substantially overlap with the training domains (Replica, ScanNet++, DL3DV). It is recognized that generalization to outdoor or large-scale environments may be limited due to mismatches in scene/depth scales. As a practical compromise, it is recommended to include additional experiments on unseen indoor/hybrid benchmarks such as TUM RGB-D, 7-Scenes, and ETH3D to more comprehensively assess the model’s robustness and effectiveness.

- The method relies on a CUT3R backbone pretrained on ScanNet v2. V2 mainly differs from V1 in labeling quality, where scenes between CUT3R’s training data and the evaluation sets used in this paper are still the same. Please justify this overlap/leakage.

**Questions:**

Please refer to the weaknesses.

---

> ### Author Response · Authors · 2025-11-25
> **Response to Reviewer 91zN(part 1/2)**
>
> We thank the reviewer for the constructive feedback. In this response, we have carefully addressed the concerns regarding novelty, runtime analysis, and data separation, and provided additional experimental results to support our claims.
>
> ### **W1: Clarification on FPS calculation protocol and detailed runtime breakdown**
>
> We thank the reviewer for this opportunity to clarify our evaluation protocol.
>
> **1. FPS and Comparison Protocol**
> We confirm that our reported FPS is calculated as $\frac{\text{Total Frames}}{\text{Total Runtime}}$, the end-to-end FPS. **This metric accounts for all system components**, including frontend inference, backend map refinement, loop closure, pose graph optimization (PGO), and other overheads.
>
> **2. Runtime Breakdown and Efficiency**
> Flash-Mono operates using two parallel threads: a Frontend thread (tracking and loop closure) and a Backend thread (mapping and refinement). As detailed in the breakdown below, the Backend is the bottleneck.
>
> | Thread | Module | Time (ms) | Note |
> | :--- | :--- | :--- | :--- |
> | **Frontend** | Feedforward Inference | 62 | Per frame |
> | | Loop Closure Detection | 3 | Per frame |
> | | **Total** | **65** | Per frame  |
> | | Loop Frame Feedforward | 62 | Per loop closure |
> | | PGO (Sim3 Optimization) | 32 | Per loop closure |
> | **Backend** | Merge & Voxelization | 0.5 | Per frame |
> | | Refine | 77 |Per frame |
> | | **Total** | **77.5** |Per frame  |
> | | GS Correction | 2 | Per loop closure |
>
> *Note: Loop closures are sparse events that typically occur every 300-400 frames, varying by dataset.*
>
> We added the FPS calculation protocl and Runtime breakdown to Appendix E.
>
> ### **W2: Clarification on Novelty and Supporting Experiments**
>
> **Novelty**
>
> We highlight two primary contributions:
> 1.  We introduce the **Feed-Forward Gaussian Splatting** and **"Predict-and-Refine" paradigm** to the field of GS-SLAM, achieving a real-time and practical GS-SLAM system that was previously unattainable. This contribution is substantiated by our ability to achieve SOTA tracking accuracy and rendering quality with 10x higher FPS than existing methods.
> 2.  At the module level, we propose a novel and highly effective **hidden-state-based loop closure mechanism**. In Ablation (c) (Figure 5), we compared our approach against both a "no loop closure" setting and a traditional "PnP+RANSAC" mechanism. The results demonstrate that our hidden-state-based method outperforms both baselines by a significant margin.
>
> **Add Ablation on Predict-and-Refine vs. Standard Local BA**
>
> Regarding the request to compare our method against "Standard Local BA," we clarify that the baseline **MonoGS** serves as the representative method for this approach, as it relies on optimizing poses and Gaussians from scratch using photometric loss (Bundle Adjustment). As detailed in Table 1 (Tracking) and Table 2 (Mapping), our method outperforms the Standard Local BA approach (represented by MonoGS) on most metrics while being nearly 20 times faster.
>
> **Add Ablation on Hidden State vs. No Hidden State**
>
> Regarding removing Hidden State completely from the frontend, this architecture is effectively represented by **MASt3R-SLAM**, which utilizes a feed-forward model without a persistent hidden state to infer the current pose solely against the previous frame. As shown in Table 1, we surpass MASt3R-SLAM on most benchmarks. Regarding removing Hidden State from the loop closure module, we addressed this in **Figure 5(c)**. The "PnP" baseline in this figure represents the traditional geometric relocalization approach (i.e., without utilizing the hidden state). The results confirm that our Hidden-State loop closure significantly reduces trajectory error compared to the "No Hidden State" baseline.

---

> ### Author Response · Authors · 2025-11-25
> **Response to Reviewer 91zN(part 2/2)**
>
> ### **W3: Add experiments on outdoor or large-scale environments**
>
> We have added new experiments on an outdoor dataset kitti. Note that other baselines such as MonoGS and DepthGS are designed for indoor scene, and fail most of the time due to high scale variance and dynamics typical of outdoor scenes. So we primarily compared with S3PO-GS, the GS-SLAM system featuring outdoor scenario. The result shows that our method is robust and generalize well to outdoor scene.
>
> | ATE RMSE[m]↓ | 0 | 5 | 6 | 7 | 8 | 28 |
> | :--- | :---: | :---: | :---: | :---: | :---: | :---: |
> | **Ours** | **12.85** | **16.58** | **9.93** | **12.08** | **45.25** | **16.75** |
> | **S3PO-GS** | 32.49 | 34.76 | 16.43 | fail | 64.74 | 23.64 |
>
> | Method | Metric | 0 | 5 | 6 | 7 | 8 | 28 |
> | :--- | :--- | :---: | :---: | :---: | :---: | :---: | :---: |
> | **S3PO-GS** | psnr ↑ | 16.65 | 15.64 | 13.5539 | fail | **17.25** | 15.3 |
> | | ssim ↑ | 0.5409 | 0.532 | 0.4726 | fail | 0.5912 | 0.5053 |
> | | lpips ↓ | 0.6254 | 0.6352 | 0.7241 | fail | **0.4626** | 0.6131 |
> | **Ours** | psnr ↑ | **17.41** | **17.01** | **15.13** | **17.89** | 16.12 | **17.47** |
> | | ssim ↑ | **0.6584** | **0.6278** | **0.5922** | **0.6036** | **0.6221** | **0.5633** |
> | | lpips ↓ | **0.5358** | **0.4871** | **0.5333** | **0.4854** | 0.4710 | **0.4581** |
>
> ### **W4: Clarification on CUT3R pretraining data overlap**
> We sincerely thank the reviewer for highlighting this important detail. While we initialize our backbone with CUT3R weights, our Gaussian heads are trained from scratch. Moreover, other feedforward SLAM systems in our experiments is trained on ScanNet, ensuring a fair comparison. DepthGS relies on UniDepthV2 (trained on ScanNet), and both S3PO-GS and MASt3R-SLAM use MASt3R, which is trained on Habitat (containing ScanNet data as noted in Dust3R paper). However, we agree that ScanNet should be seen as an in-domain evaluation. To assess generalization capabilities on unseen scenes (out-of-domain), we rely on the BundleFusion experiments, which confirm that our method generalizes effectively beyond the training distribution. We've added the explanation to the paper.(line 370)

---

> > ### Comment · Reviewer_91zN · 2025-11-27
> >
> > I appreciate the detailed explanation and additional experiments in the authors’ reply. After carefully reviewing all rebuttal materials, I find that my concerns have been sufficiently addressed, and I will therefore raise my score accordingly.

---

> > > ### Author Response · Authors · 2025-11-27
> > >
> > > We thank the reviewer for the time spent re-evaluating our work and for the encouraging feedback. We are glad to hear that our response satisfactorily addressed your concerns. We truly appreciate your recognition of our work.

---

### Official Review · Reviewer_5jYg · 2025-10-28

**Soundness:** 3
**Presentation:** 4
**Contribution:** 3
**Rating:** 8
**Confidence:** 4

**Summary:**

This paper presents a Gaussian-based SLAM system that adopts a feed-forward paradigm to predict the attributes of 2D Gaussians and relative poses by leveraging multi-frame contextual information. The core contribution lies in integrating a feed-forward module into the GS-based SLAM framework, effectively replacing the time-consuming optimization process used in most recent GS-based SLAM systems. In addition, a loop closure module based on the hidden states of keyframes is introduced to enhance tracking accuracy and mitigate error accumulation. Experimental results demonstrate both the effectiveness and efficiency of the proposed SLAM system.

**Strengths:**

1. The paper is well-written and easy to follow, and its motivation is clear and well-founded.
2. It introduces a novel feed-forward paradigm for Gaussian-based SLAM, replacing traditional optimization-based processes and significantly improving efficiency. By eliminating costly optimization steps, the proposed method contributes to the development of real-time and lightweight Gaussian-based SLAM frameworks.
3. The paper trains a recurrent feed-forward frontend model that aggregates multi-frame visual features into a hidden state via cross-attention and jointly predicts camera poses and per-pixel Gaussian properties. The hidden state is further utilized in the loop closure module to reduce cumulative drift and enhance tracking robustness.

**Weaknesses:**

1. Some state-of-the-art RGB-based SLAM methods, such as Photo-SLAM and DROID-Splat, are not included in the comparison. In addition, since S3PO-GS primarily focuses on outdoor scenes, it would be beneficial to include evaluations on outdoor datasets to better demonstrate the proposed SLAM system’s capability in handling long and complex outdoor sequences.
2. It would be beneficial to include a quantitative analysis of the total number of Gaussians required to represent the entire scene. This statistic could provide valuable insight into the efficiency of the proposed feed-forward prediction and map fusion modules, and demonstrate how effectively the method balances compactness with reconstruction quality.

**Questions:**

Please refer to the weakness part.

---

> ### Author Response · Authors · 2025-11-25
> **Response to Reviewer 5jYg(part 1/2)**
>
> We sincerely thank the reviewer for the positive assessment (Rating 8) of the novelty and efficiency of our feed-forward paradigm. We respond to your questions below and would appreciate it if you could let us know if our response addresses your concerns.
>
> ### **W1: Compare with S3PO-GS on Outdoor dataset and include more baselines**
> We have added new experiments on an outdoor dataset kitti. Note that other baselines such as MonoGS and DepthGS are designed for indoor scene, and fail most of the time due to high scale variance and dynamics typical of outdoor scenes. So we primarily compared with S3PO-GS, the GS-SLAM system featuring outdoor scenario. The result shows that our method is robust and generalize well to outdoor scene.
>
> | ATE RMSE[m]↓ | 0 | 5 | 6 | 7 | 8 | 28 |
> | :--- | :---: | :---: | :---: | :---: | :---: | :---: |
> | **Ours** | **12.85** | **16.58** | **9.93** | **12.08** | **45.25** | **16.75** |
> | **S3PO-GS** | 32.49 | 34.76 | 16.43 | fail | 64.74 | 23.64 |
>
> | Method | Metric | 0 | 5 | 6 | 7 | 8 | 28 |
> | :--- | :--- | :---: | :---: | :---: | :---: | :---: | :---: |
> | **S3PO-GS** | psnr ↑ | 16.65 | 15.64 | 13.5539 | fail | **17.25** | 15.3 |
> | | ssim ↑ | 0.5409 | 0.532 | 0.4726 | fail | 0.5912 | 0.5053 |
> | | lpips ↓ | 0.6254 | 0.6352 | 0.7241 | fail | **0.4626** | 0.6131 |
> | **Ours** | psnr ↑ | **17.41** | **17.01** | **15.13** | **17.89** | 16.12 | **17.47** |
> | | ssim ↑ | **0.6584** | **0.6278** | **0.5922** | **0.6036** | **0.6221** | **0.5633** |
> | | lpips ↓ | **0.5358** | **0.4871** | **0.5333** | **0.4854** | 0.4710 | **0.4581** |
>
>
> We evaluated Photo-SLAM's performance on ScanNet and BundleFusion, Photo-SLAM fails to track robustly and reinitializes multiple times. This is mainly due to the feature-based frontend tracking method it adopted, as feature-based methods are prune to fast motion causing motion blur. The PSNR is usually below 12, as photo-SLAM does not explicitly handle tracking failure. After Photo-SLAM loses track, it resets the pose of the next frame to the origin and continues the mapping process, resulting in a stacked map from different time. We here report the tracking metrics of Photo-SLAM. We also added discussion of some recent GS-SLAM systems(SEGS-SLAM Droid-Splat WildGS-SLAM) to related work.
>
> | ATE [cm] ↓ | ScanNetV1 | | | | | | BundleFusion | | | | |
> | :--- | :---: | :---: | :---: | :---: | :---: | :---: | :---: | :---: | :---: | :---: | :---: |
> | **Method** | **0054** | **0059** | **0106** | **0169** | **0233** | **0465** | **apt0** | **apt2** | **copyroom** | **office0** | **office2** |
> | ORB-SLAM3 | 243.26 | 90.67 | 178.13 | 60.15 | 25.01 | 181.86 | 87.37 | 265.64 | 27.60 | 116.33 | 49.33 |
> | DROID-SLAM | 161.22 | 69.92 | 89.11 | 28.26 | 74.01 | 117.27 | 89.38 | 148.04 | 19.71 | 31.41 | 73.91 |
> | MonoGS | 70.19 | 97.24 | 150.89 | 191.98 | 62.45 | 113.19 | 122.59 | 142.54 | 53.41 | 62.67 | 127.02 |
> | DepthGS | 192.18 | 93.69 | 140.19 | 205.92 | 81.90 | 121.01 | 67.52 | 119.74 | 14.59 | 40.42 | 16.05 |
> | S3PO-GS | 69.36 | 16.52 | 26.15 | 87.04 | 27.09 | 96.35 | 92.49 | 97.90 | 21.88 | 64.22 | 69.88 |
> | MASt3R-SLAM | 13.25 | 10.89 | 15.83 | 15.24 | **10.99** | 15.74 | **9.65** | 13.66 | 9.28 | 9.97 | 9.92 |
> | Ours | **11.69** | **8.89** | **10.83** | **10.16** | 12.13 | **13.00** | 11.44 | **12.36** | **7.34** | **8.74** | **9.34** |
> | photo-SLAM | 332.03 | 205.01 | 359.85 | 151.61 | 195.72 | 294.20 | 247.19 | 320.91 | 54.03 | 271.87 | 298.92 |

---

> ### Author Response · Authors · 2025-11-25
> **Response to Reviewer 5jYg(part 2/2)**
>
> ### **W2: Quantitative analysis of the total number of Gaussians**
>
> We provide a quantitative comparison of the Gaussian count on the TUM dataset below. The data of baseline methods is sourced from CaRtGS.
>
> | Gaussian Count| fr1/desk | fr2/xyz | fr3/office |
> | :--- | :--- | :--- | :--- |
> | MonoGS | 26.64k | 43.59k | 35.24k |
> | Photo-SLAM | 40.00k | 0.10m | 81.16k |
> | SplaTAM | 0.96m | 6.36m | 0.79m |
> | Gaussian-SLAM | 0.76m | 0.69m | 1.47m |
> | GS-ICP-SLAM | 0.53m | 1.91m | 2.09m |
> | Ours | 0.63m | 0.98m | 0.61m |
>
> As shown in the table, our method maintains a moderate level of map compactness compared to existing approaches. We anticipate that future advancements in feed-forward models capable of generating pixel-unaligned Gaussians will further enhance storage efficiency.
>
> Regarding MonoGS, while it achieves the highest map compactness, we observed that it frequently encounters Out-Of-Memory (OOM) failures on long sequences (exceeding 5,000 frames) in our experiments. Upon analyzing their implementation, we attribute this instability mainly to the memory overhead of the covisibility index required for their pruning mechanism. This indicates that MonoGS's final map compactness comes at the cost of runtime memory stability during large-scale reconstruction.

---

> > ### Comment · Reviewer_5jYg · 2025-11-28
> >
> > I appreciate the authors’ detailed response, including the comprehensive comparisons and analyses. My original concerns have been fully addressed. The paper presents an interesting idea, and the experimental results convincingly demonstrate the effectiveness of the proposed methods. I will therefore maintain my original positive rating.

---

> ### Author Response · Authors · 2025-11-28
>
> We sincerely thank you for your prompt feedback and for confirming that our response has fully addressed your original concerns. We deeply appreciate your recognition of our work and are open to further discussions.

---

### Official Review · Reviewer_eTqG · 2025-10-31

**Soundness:** 3
**Presentation:** 3
**Contribution:** 3
**Rating:** 6
**Confidence:** 4

**Summary:**

This paper presents Flash-Mono, a monocular SLAM system that integrates a feed-forward model with 2D Gaussian splatting (2DGS) for real-time scene reconstruction and camera tracking. The key idea is to replace the traditional per-frame optimization of Gaussian attributes with a recurrent network that directly predicts poses and Gaussians from sequential inputs. The proposed system consists of three modules: a feed-forward frontend, a 2DGS-based mapping backend, and a loop closure mechanism based on hidden states. The authors claim a 10× speedup over existing GS-SLAM methods while achieving state-of-the-art performance in tracking and rendering quality on ScanNet and BundleFusion datasets.

**Strengths:**

The combination of a recurrent feed-forward model with 2DGS for monocular SLAM is novel. The use of hidden states as submap descriptors for loop closure is creative.

The method achieves strong results in both tracking (ATE) and rendering (PSNR, SSIM, LPIPS), outperforming recent GS-SLAM systems.

The paper is well-organized and easy to follow.

**Weaknesses:**

The evaluation is limited to indoor datasets (ScanNet, BundleFusion). It is unclear how the method generalizes to outdoor or large-scale environments.

While the hidden state mechanism is innovative, its capacity for long-term consistency is not deeply analyzed.

The paper does not discuss model size or memory usage, which are important for deployment on resource-constrained devices.

**Questions:**

1. How does the method perform in outdoor or large-scale scenes where scale variation and dynamics are more challenging?
﻿
2. Could the hidden state be further exploited for lifelong mapping or incremental learning beyond submap-based reset?
﻿
3. Have you considered comparing with non-GS SLAM systems in terms of robustness under motion blur or low-texture scenes?
﻿
4. What is the memory footprint of the model, and is it feasible for mobile or embedded platforms?

---

> ### Author Response · Authors · 2025-11-25
> **Response to Reviewer eTqG(part 1/3)**
>
> Thank you for your constructive comments and suggestions. We found that your suggestion on analyzing the hidden state's potential for life-long mapping particularly insightful. We respond to your questions below and would appreciate it if you could let us know if our response addresses your concerns.
>
> ### **W1&Q1: Include evaluation on outdoor or large-scale dataset to show the method's performance under challenging scale variation and dynamics**
> We have added new experiments on an outdoor dataset kitti. Note that MonoGS and DepthGS are designed for indoor scene, and fail most of the time due to high scale variance and dynamics typical of outdoor scenes. So we primarily compared with S3PO-GS, the GS-SLAM system featuring outdoor scenario. The result shows that our method is robust and generalize well to outdoor scene.
>
> | ATE RMSE[m]↓ | 0 | 5 | 6 | 7 | 8 | 28 |
> | :--- | :---: | :---: | :---: | :---: | :---: | :---: |
> | **Ours** | **12.85** | **16.58** | **9.93** | **12.08** | **45.25** | **16.75** |
> | **S3PO-GS** | 32.49 | 34.76 | 16.43 | fail | 64.74 | 23.64 |
>
> | Method | Metric | 0 | 5 | 6 | 7 | 8 | 28 |
> | :--- | :--- | :---: | :---: | :---: | :---: | :---: | :---: |
> | **S3PO-GS** | psnr ↑ | 16.65 | 15.64 | 13.5539 | fail | **17.25** | 15.3 |
> | | ssim ↑ | 0.5409 | 0.532 | 0.4726 | fail | 0.5912 | 0.5053 |
> | | lpips ↓ | 0.6254 | 0.6352 | 0.7241 | fail | **0.4626** | 0.6131 |
> | **Ours** | psnr ↑ | **17.41** | **17.01** | **15.13** | **17.89** | 16.12 | **17.47** |
> | | ssim ↑ | **0.6584** | **0.6278** | **0.5922** | **0.6036** | **0.6221** | **0.5633** |
> | | lpips ↓ | **0.5358** | **0.4871** | **0.5333** | **0.4854** | 0.4710 | **0.4581** |
>
> We observed that in the initial segments of each sequence, the tracking performance of S3PO-GS and our system is comparable. **As the sequence length increases, our method demonstrates superiority in scale consistency compared to S3PO-GS.** This is primarily attributed to our hidden state mechanism, which effectively exploits multi-frame information to predict scale more accurately, thereby demonstrating the robustness of our system in large-scale environments. Despite these promising results, **it is foreseeable that in extremely complex, large-scale environments (e.g., shopping malls), the current system may still face challenges regarding global map memory pressure and loop closure accuracy**. To address this, more engineering efforts could be paid into **extending our framework to a hierarchical architecture(map and loop closure).** We believe our existing submap descriptor can serve as a starting point for this hierarchical partitioning of global map.
>
> **For performance under extremely dynamic scenes, you can refer to the case study in Appendix G Figure 9** It demonstrates the model's robustness to extreme environmental changes, such as illumination variations (day/night) and structural changes (moving people and furniture).

---

> ### Author Response · Authors · 2025-11-25
> **Response to Reviewer eTqG(part 2/3)**
>
> ### **W2 & Q2: Analyze the hidden state mechanism's capacity for long-term consistency and life-long mapping.**
> We thank the reviewer for provioding this valuable insight. Currently, the hidden state is primarily used for loop closure relocalization to improve in-session long-term consistency. Ablation (c) showed its superioty to traditional methods. For life-long mapping, which is beyond the scope of our original paper, we believe the hidden state mechanism can also be extended to handle environmental dynamics and maintain map consistency.
>
> To define the problem, we refer to Zhao et al. [2], who state that **a lifelong mapping system should "construct and maintain an up-to-date map while it continuously operates in a changing environment."** This definition implies two non-trivial challenges: (1) Robustly localizing within a changed scene (Operating), and (2) Synchronizing the map representation over time (Maintaining).
>
> **1. Relocalizing new observation against outdated map**
> First, the system must be capable of relocalizing an observation of the new environment to the old outdated Map. Our hidden state mechanism can be used to achieve this in the exact same way as described in Sec 4.3(Loop Frame Feed-Forward for Relocalization). **To validate this point, we conducted a case study in Appendix. G Figure 9** to show our feedforward model is able to relocalize robustly even when the environment changed severly. We first input 8 frames of a room taken during night with the curtains closed and seat back in place into the model to generate a hidden state. And then we input the model with an observation of the same scene during the day with the curtains open and a man sitting on the chair. The result shows our model robustly relocalized the new frame against the old map and predicted correct geometry. We believe training specifically to changed environment dataset will further improve this robustness.
>
> **2. Updating the Map Representation**
> Second, the system must update the map representation with new observations to stay synchronized with the current environment. Our system employs three representations: the Gaussian Map, the Pose Graph, and the Hidden State (which acts as a compact submap descriptor). While updating Gaussian primitives[1] and pose graphs[2] is well-studied, updating the hidden state descriptor is non-trivial.
>
> Simply continuing to aggregate new observations into a hidden state eventually leads to saturation and catastrophic forgetting. To address this for life-long mapping, we propose two potential strategies:
>
> 1. **Discrete State Replacement:** We can adopt a simple replacement strategy. When significant environmental change is detected during relocalization, rather than modifying the old hidden state, we can generate a *fresh* hidden state $M_{new}$ from the current observation. We then either replace the obsolete state in the Bag or append the new state as an alternative descriptor for that physical location (handling "day vs. night" variations).
> 2. **design better hidden state update strategy:** Instead of discrete replacement, we can implement a continuous update strategy inspired by TTT3R [3]. This approach reframes the hidden state update as an online learning problem, treating the state as "fast weights" optimized via gradient descent during inference. Crucially, TTT3R introduces a confidence-guided update rule, where the learning rate is dynamically derived from the alignment between the current memory state and the incoming observation. This acts as a self-supervised gating mechanism: it allows the hidden state to selectively integrate persistent environmental changes (where alignment confidence is high) while suppressing transient noise or inconsistent updates (where confidence is low). By adopting this formulation, our system can evolve to capture gradual domain shifts—such as seasonal changes or furniture rearrangement—while mitigating the catastrophic forgetting typically associated with recurrent updates, ensuring the map remains both up-to-date and geometrically consistent over the long term.
>
> We addded the discussion of the system's potential in life long mapping to Appendix G of the paper. We believe this provide valuable future direction of our system.
>
> **[1] GS-LTS: 3D Gaussian Splatting-Based Adaptive Modeling for Long-Term Service Robots**
>   Bin Fu, Jialin Li, Bin Zhang, Ruiping Wang, Xilin Chen.
>   *arXiv preprint arXiv:2503.17733*, 2025.
>
> **[2] A general framework for lifelong localization and mapping in changing environment**
>   Min Zhao, Xin Guo, Le Song, Baoxing Qin, Xuesong Shi, Gim Hee Lee, Guanghui Sun.
>   In *2021 IEEE/RSJ International Conference on Intelligent Robots and Systems (IROS)*, pp. 3305-3312. IEEE, 2021.
>
> **[3] TTT3R: 3D Reconstruction as Test-Time Training**
>   Xingyu Chen, Yue Chen, Yuliang Xiu, Andreas Geiger, Anpei Chen.
>   *arXiv preprint arXiv:2509.26645*, 2025.

---

> ### Author Response · Authors · 2025-11-25
> **Response to Reviewer eTqG(part 3/3)**
>
> ### **W3 & Q4: Discuss model size or memory usage to assess feasibility of deployment on resource constrained devices**
>
> **1. model size**
>
> The size of our model is detailed in the table below.
>
> | Component | Total Parameters |
> | :--- | :--- |
> | Encoder | 303.1 M |
> | Decoder | 380.8 M |
> | Heads & Tokens | 111.8 M |
> | Total | 795.7 M |
>
> We've added the discussion of model size to Appendix C.1.
>
> **2. Acceleration and analyzation on deployment for resource-constrained platform**
>
> As detailed in Appendix C.2, we used a laptop version 4060 GPU(8GB) to simulate resource-constrained platform relative to our experimental setup. **We performed CUDA Graph Optimization and parameter quantification** on the model to adapt to the low computation capability. Result shows that we achieve approximately **2 times memory compression and 3.3 times inference time acceleration**. The inference time on laptop 4060(8G) after acceleration is just slightly slower than the inference time on 4090 GPU(24G) used in our experiments setup. (83ms vs. 62ms).
>
> We project that with further engineering, our system can achieve real-time performance on the NVIDIA Jetson AGX Orin. Consequently, our system is **feasible for autonomous driving** (a primary application scenario for Gaussian Splatting map representations) and **service robotics**. However, we acknowledge that **deployment on strictly power-limited platforms, such as drones, remains challenging** at this stage. Notably, existing GS-SLAM systems like Photo-SLAM [1] also view hardware comparable to the Jetson AGX Orin as the least requirement, explicitly stating that it "can run at real-time speed using an embedded platform such as Jetson AGX Orin, showing the potential of robotics applications."
>
>
>
> ### **Q3: Have you considered comparing with non-GS SLAM systems in terms of robustness under motion blur or low-texture scenes?**
>
> Yes, we have compared our method against leading non-GS SLAM systems. We evaluate absolute trajectory estimation error in Table 1 across different methods on Scannet and BundleFusion. The ScanNet dataset is characterized by severe motion blur and diverse lighting conditions. Our method surpass feature-based(ORB-SLAM3) and learning-based(Droid-SLAM, Mast3r-SLAM) baselines. We believe this advantage is mainly due to our model which leverages large-scale training across diverse datasets to achieve greater robustness and our novel loop closure mechanism that is tightly coupled with the model architecture.
>
> [1] Photo-SLAM: Real-time Simultaneous Localization and Photorealistic Mapping for Monocular, Stereo, and RGB-D Cameras. Huajian Huang, Longwei Li, Hui Cheng, Sai-Kit Yeung. In Proceedings of the IEEE/CVF Conference on Computer Vision and Pattern Recognition (CVPR), 2024.

---

> > ### Comment · Reviewer_eTqG · 2025-11-28
> >
> > Thank the authors for their comprehensive rebuttal. The additional experiments and the clarifications on the system's potential for lifelong mapping have effectively addressed my main concerns.
> > I am satisfied with the response and support the paper's acceptance.

---

> ### Author Response · Authors · 2025-11-28
>
> We are so glad to hear that our clarifications have successfully addressed your main concerns. We sincerely appreciate your constructive feedback, which has improved the quality of our manuscript. We remain available to answer any further questions or concerns you may have. We will continue to refine our paper.

---

### Official Review · Reviewer_S5jM · 2025-10-31

**Soundness:** 2
**Presentation:** 2
**Contribution:** 2
**Rating:** 2
**Confidence:** 4

**Summary:**

The paper introduces Flash-Mono, an monocular 3D Gaussion Splatting SLAM that employs the feed-forward paradigm to improve the accuracy and speed of GS-SLAM methods. The propose method trains a recurrent feed-forward fronted to predicts local camera posed and per-pixel 2D Gaussions and cobmines it with backend global mapping and loop closure to achieve final tracking and mapping results. The experimental results demonstrate the superiority of the proposed Flash-Mono.

**Strengths:**

The authors introduce a dedicated feed-forward Gaussion prediction based SLAM framework with sophisticated frontend and backend component designs to improve tracking and mapping performance in various scenes. The designs are reasonable and well-supported by and experimental results.

**Weaknesses:**

1.The feed-forward reconstruction-based SLAM is new direction for SLAM community, the authors should conduct extensive experiments to show its superiority and weakness on various SLAM scenes. That is, the authors should provide more results on indoor and outdoor benchmarks and mapping reconstruction metric like Completion and Chamfer.

2.The authors do not provide a sufficient comparison with other feed-forward SLAM method, e.g., VGGT-SLAM. Additionally, the ablation results are too simple and no failure analysis about the limitations of the proposed method.

3.The explanation of some technical details is unclear. What data is used to train the frontend network, how the training data influence the final SLAM results?

**Questions:**

1.When the backend mapping fusion and refinement correction are performing, what frequency does the front-end and backend interacts and how it influence the final results.

2.How the historical interpretation P_a^j is calculated, project point cloud from current frame I_j to historical submap C_a?

3.Does the pose relocalization process perform after the local map refinement or before, what is the detailed pipeline of the whole system?

4.In Loop Correction of Gaussian Map, does this simple transformation of 2DGS primitives generate bad overlap between 2DGS, resulting in bad rendering and mapping.

---

> ### Author Response · Authors · 2025-11-25
> **Response to Reviewer S5jM(part 1/4)**
>
> We appreciate the constructive comments. We have performed extensive additional experiments and provided detailed explanation to the technical questions. We would appreciate it if you could let us know if our response addresses your concerns.
>
> ### **W1.1: More results on indoor and outdoor benchmarks**
> To address the reviewer's concern, in addition to the two indoor datasets in the original paper, we have conducted new experiments on the outdoor dataset KITTI.  We respectfully note that other baselines, such as MonoGS and DepthGS, are designed primarily for small-scale indoor environments. In our tests on KITTI, these methods encountered tracking failures or crashed due to the large scale variance typical of outdoor driving scenarios. So we primarily compared our method against S3PO-GS, which features outdoor scene.
>
> | ATE RMSE[m]↓ | 0 | 5 | 6 | 7 | 8 | 28 |
> | :--- | :---: | :---: | :---: | :---: | :---: | :---: |
> | **Ours** | **12.85** | **16.58** | **9.93** | **12.08** | **45.25** | **16.75** |
> | **S3PO-GS** | 32.49 | 34.76 | 16.43 | fail | 64.74 | 23.64 |
>
> | Method | Metric | 0 | 5 | 6 | 7 | 8 | 28 |
> | :--- | :--- | :---: | :---: | :---: | :---: | :---: | :---: |
> | **S3PO-GS** | psnr ↑ | 16.65 | 15.64 | 13.5539 | fail | **17.25** | 15.3 |
> | | ssim ↑ | 0.5409 | 0.532 | 0.4726 | fail | 0.5912 | 0.5053 |
> | | lpips ↓ | 0.6254 | 0.6352 | 0.7241 | fail | **0.4626** | 0.6131 |
> | **Ours** | psnr ↑ | **17.41** | **17.01** | **15.13** | **17.89** | 16.12 | **17.47** |
> | | ssim ↑ | **0.6584** | **0.6278** | **0.5922** | **0.6036** | **0.6221** | **0.5633** |
> | | lpips ↓ | **0.5358** | **0.4871** | **0.5333** | **0.4854** | 0.4710 | **0.4581** |
>
> The results demonstrate that Flash-Mono generalizes well to outdoor settings.
>
> ### **W1.2: More Mapping Geometric Metrics**
> We appreciate the reviewer’s insight. Most existing GS-SLAM (e.g., MonoGS , S3PO-GS ) rely on rendering metrics (PSNR, SSIM, LPIPS) to assess mapping quality. **We highly agree with you that high-quality rendering does not gurantee accurate geometric reconstruction.**
>
> We recognize that Completion and Chamfer Distance are the conventional metrics for evaluating geometric accuracy in pointcloud-based SLAM. However,  Gaussian Maps is a sparse representation, where a single primitive is often optimized to represent a large surface area. Consequently, the sparse distribution of Gaussian centroids fails to reflect the actual geometric coverage. Thus, metrics designed for pointcloud like **Chamfer Distance and Completion is unreliable for evaluating Gaussian Maps.** To address the reviewer's concern regarding geometric quality, we **employ Scale-aligned Depth L1 Error** as a proxy metric in the initial submission. This metric calculates the absolute difference between the depth rendered from the Gaussian map and the ground truth, incorporating global scale alignment to account for monocular scale ambiguity. Since the rendering process of depth is native to Gaussian, Depth L1 is an effective metrics to evaluate the geometric quality of Gaussian Map.
>
>
>
> | Scale-aligned Depth L1 [m] $\downarrow$ | ScanNetV1 | | | | | | BundleFusion | | | | |
> | :--- | :---: | :---: | :---: | :---: | :---: | :---: | :---: | :---: | :---: | :---: | :---: |
> | Methods | 0054 | 0059 | 0106 | 0169 | 0233 | 0465 | apt0 | apt2 | copyroom | office0 | office2 |
> | MonoGS | 1.06 | 1.27 | 1.41 | 1.56 | 0.89 | 0.82 | 0.96 | 1.28 | 1.18 | 1.15 | 1.26 |
> | DepthGS | 0.45 | 0.66 | 0.52 | 0.48 | 0.41 | 0.47 | 0.37 | **0.29** | 0.13 | 0.18 | 0.21 |
> | S3PO-GS | 0.58 | 0.35 | 0.55 | 0.66 | 0.28 | 0.89 | 0.72 | 0.99 | 0.41 | 1.01 | 0.85 |
> | Ours | **0.16** | **0.23** | **0.51** | **0.17** | **0.35** | **0.44** | **0.33** | 0.35 | **0.11** | **0.11** | **0.18** |
>
>
> As demonstrated in the table, our method achieves best geometric accuracy. This confirms that Flash-Mono succeeds in building maps that are not only visually plausible but also geometrically accurate.

---

> ### Author Response · Authors · 2025-11-25
> **Response to Reviewer S5jM(part 2/4)**
>
> ### **W2: comparison with more Feedforward SLAM methods (e.g., VGGT-SLAM)**
>
> We thank the reviewer for pointing out this relevant work. In our current evaluation, we primarily compared against feedforward methods including MASt3R-SLAM, DepthGS, S3PO-GS. **We did not include VGGT-SLAM in the initial submission mainly due to time constraints**, as VGGT-SLAM is accepted to NIPS one week prior to the submission deadline of ICLR2026.
>
> Regarding VGGT-SLAM, while we recognize its contributions and strengths, we respectfully suggest that **a direct comparison might be unfair due to the following 2 reasons**:
> 1.  **High latency unacceptable for typical SLAM scenario**
>     Conventional SLAM methods process frames strictly sequentially. In contrast, VGGT-SLAM inherits the batch-processing nature of its VGGT, introducing a latency of ~100 frames to generate poses. This is unacceptable for typical SLAM usage that require online pose estimation. Also, this batch approach effectively utilizes "future information" to refine current estimates.
> 2.  **Performance Gains Mainly from VGGT**
>     As noted in VGGT-SLAM's own experiment, the performance of their proposed system is on par with a naive combination of VGGT + traditional SE3 loop closure. This suggests that its core advantage stems largely from the VGGT model rather than the SLAM pipeline they design.
>
> Nevertheless, **we outperformed VGGT-SLAM on approximately half of the trajectory evaluation.** This indicates that our novel hidden state-based loop closure mechanism successfully bridges the performance gap caused by the unequal comparison conditions.
>
> | ATE [cm] ↓ | ScanNetV1 | | | | | | | BundleFusion | | | | |
> | :--- | :--- | :---: | :---: | :---: | :---: | :---: | :---: | :---: | :---: | :---: | :---: | :---: |
> | Category | Method | 0054 | 0059 | 0106 | 0169 | 0233 | 0465 | apt0 | apt2 | copyroom | office0 | office2 |
> | **Online** | ORB-SLAM3 | 243.26 | 90.67 | 178.13 | 60.15 | 25.01 | 181.86 | 87.37 | 265.64 | 27.60 | 116.33 | 49.33 |
> | | DROID-SLAM | 161.22 | 69.92 | 89.11 | 28.26 | 74.01 | 117.27 | 89.38 | 148.04 | 19.71 | 31.41 | 73.91 |
> | | MonoGS | 70.19 | 97.24 | 150.89 | 191.98 | 62.45 | 113.19 | 122.59 | 142.54 | 53.41 | 62.67 | 127.02 |
> | | DepthGS | 192.18 | 93.69 | 140.19 | 205.92 | 81.90 | 121.01 | 67.52 | 119.74 | 14.59 | 40.42 | 16.05 |
> | | S3PO-GS | 69.36 | 16.52 | 26.15 | 87.04 | 27.09 | 96.35 | 92.49 | 97.90 | 21.88 | 64.22 | 69.88 |
> | | MASt3R-SLAM | 13.25 | 10.89 | 15.83 | 15.24 | **10.99** | 15.74 | **9.65** | 13.66 | 9.28 | 9.97 | 9.92 |
> | | Ours | **11.69** | **8.89** | **10.83** | **10.16** | 12.13 | **13.00** | 11.44 | **12.36** | **7.34** | **8.74** | **9.34** |
> | **Semi-Online**| VGGT-SLAM | 12.66 | 6.74 | 11.46 | 8.16 | 11.89 | 13.23 | 9.23 | 13.20 | 6.89 | 7.23 | failed |
>
>
>
> ### **W3: What data is used to train the frontend network, how the training data influence the final SLAM results?**
> We agree that these details are fundamental to the method's success. As stated in Section 4.1 and Appendix D, we trained the frontend model on indoor/outdoor datasets including ScanNet++, DL3DV and Replica with a three-stage learning curriculum. This diverse training dataset is key to the generalization and robustness of our system.

---

> ### Author Response · Authors · 2025-11-25
> **Response to Reviewer S5jM(part 3/4)**
>
> ### **Q3.1: What is the detailed pipeline of the system?**
> To provide a clearer overview, our pipeline can be divided into two operational modes: the Standard Tracking & Mapping (Per-Frame) and the Loop Closure (Event-Triggered).
>
> 1.  **Standard Tracking & Mapping (Per-Frame):**
>     For every incoming frame, the frontend performs **model inference** to jointly predict the camera pose and the local Gaussian point cloud. These predictions are transmitted to the backend, which executes **Map Fusion** to integrate the new Gaussians into the global map, followed by **Local Refinement** to polish the geometry.
>
> 2.  **Loop Closure (Event-Triggered):**
>     When the frontend detects a revisit to a historical area (a sparse event), it triggers a specialized workflow. We first perform **Pose Relocalization** via a single forward pass (inferring the current pose relative to the historical submap). The relocalization information is used in a **Sim(3) Pose Graph Optimization (PGO)** to correct accumulated trajectory drift. Finally, the corrected global poses are sent to the backend to execute **Loop Correction of Gaussian Map**, rigidly warping the Gaussian map to align with the updated trajectory.
>
> For more details, please refer to Sec 4.
>
> ### **Q3.2: Does the pose relocalization process perform after the map refinement or before?**
>
> **There is no sequential dependency.** Map Refinement is a standard operation performed for every keyframe, whereas Pose Relocalization is only triggered by occassional loop closure events. However, in the specific case where a loop closure is identified, the execution order is:
>
> 1.  The frontend model inferences on the current frame to generate pose and local Gaussian Map of the current frame, the data is then pushed to the backend;
> 2.  The backend receives the data and starts Map Fusion & Refinement of the current frame;
> 3.  A loop closure event is detected in the frontend (i.e. current frame observes the same area with a historical frame);
> 4.  Pose Relocalization & Pose Graph Optimization is performed in the frontend to correct the trajectory, the updated poses is pushed to backend;
> 5.  The backend receives the updated poses, if Map Fusion & Refinement has finished, starts loop correction of Gaussian Map immediately, otherwise wait for the Map Fusion & Refinement to complete.
>
> ### **Q1: When the backend mapping fusion and refinement correction are performing, what frequency does the front-end and backend interacts and how it influence the final results.**
> For mapping fusion and refinement, data is passed from frontend to backend every frame；for loop correction of Gaussian Map, data is passed from frontend to backend every detected loop closure. This frequent interaction does not create a bottleneck. This high frequency of exchanging data ensures the global map remains "fresh" at all times,maximizing the robustness and accuracy of the SLAM system.
> ### **Q2: How the historical interpretation $\mathcal{P}_a^j$ is calculated, project point cloud from current frame $I_j$ to historical submap $C_a$?**
>
> This is the core innovation in our loop closure module and is detailed in Sec 4.1 (Loop Frame Feed-Forward for Relocalization). Not by projection, but by a single forward pass of current frame $I_j$ conditioned on past hidden state $M_a$ of submap $C_a$. More specifically:
> 1.  We retrieve the cached hidden state $M_a$ from the "Bag of Hidden States". This state contains the geometric context of the historical submap $C_a$.
> 2.  We feed the current image $I_j$ into the model, conditioned on this frozen historical state $M_a$ (i.e., $f(I_j, M_a)$).
> 3.  The model then predicts the historical interpretation of the current view, $\mathcal{P}_a^j$.

---

> ### Author Response · Authors · 2025-11-25
> **Response to Reviewer S5jM(part 4/4)**
>
> ### **Q4: In Loop Correction of Gaussian Map, does this simple transformation of 2DGS primitives generate bad overlap between 2DGS, resulting in bad rendering and mapping.**
>
> We thank the reviewer for this highly insightful observation. Applying a rigid transformation (Sim(3)) to Gaussian primitives will disrupt the delicate Gaussian attributes optmizied over time, leading to a temporary drop in rendering metrics immediately after loop closure.
>
> However, we argue that this trade-off is necessary: The high rendering metrics prior to loop closure were a result of 3DGS **overfitting to incorrect camera poses**. While the image looked good locally, the global geometry was distorted. The drop in metrics after loop closure reflects the system transition from a locally overfitted state to a globally geometrically consistent state. Also, as the system continues to run, our backend refinement quickly adapts the Gaussian parameters to the corrected poses, recovering the rendering quality while maintaining the correct trajectory.
>
> The reviewer’s question points toward a open problem for the community. We believe a possible solution can be extending **Non-rigid Deformation Fields** (as pioneered by *Kintinuous* [1] for point clouds) to anisotropic Gaussian primitives. A future direction is to mathematically derive a deformation field that maintains the intricate relationship of neighboring Gaussians.
>
> [1]: Whelan, T., Kaess, M., Leonard, J. J., & McDonald, J. (2013). Deformation-based loop closure for large scale dense RGB-D SLAM. *2013 IEEE/RSJ International Conference on Intelligent Robots and Systems*, 548-555.

---

> > ### Comment · Reviewer_S5jM · 2025-11-28
> >
> > I thank the authors for their significant effort in rebuttal and conducting extra experiments. After reading the rebuttal, I will raise my previous rating. However, I think this work is promising, and I would raise my score accept if the authors could add the comparison results of traditional SLAM method (even if the proposed method does not outperform them) in the rebuttal.

---

> > > ### Author Response · Authors · 2025-11-28
> > >
> > > We sincerely thank the reviewer for the prompt feedback and the encouraging decision to raise the score based on our rebuttal and the potential to raise the rating further if more comparison are provided. We are currently conducting experiments for the comparison with traditional baselines. We will update the thread with these additional results later.

---

> > > ### Author Response · Authors · 2025-12-02
> > > **Add the comparison results of traditional SLAM method**
> > >
> > > In the original paper, **we have compared with preious SOTA traditional (non-GS) baselines including feature-based ORB-SLAM3, learning-based DROID-SLAM and Mast3r-SLAM(see table 1).**  It is important to note that these traditional methods utilize point-based representations and inherently can't render RGB images from novel views, making them incomparable on photorealistic metrics (PSNR, SSIM, LPIPS). To fully address the reviewer's interest in baseline comparisons and to demonstrate the superiority of our rendering quality, we have gone a step further. Instead of limiting the comparison to traditional methods, **we added a comparison with GO-SLAM, the current SOTA in NeRF-based SLAM which supports view synthesis.** As shown in the table below, our method significantly outperforms GO-SLAM across all scenes:
> > >
> > > | Method | Metric | 0054 | 0059 | 0106 | 0169 | 0233 | 0465 | apt0 | apt2 | copyroom | office0 | office2 |
> > > | :--- | :--- | :---: | :---: | :---: | :---: | :---: | :---: | :---: | :---: | :---: | :---: | :---: |
> > > | **GO-SLAM** | SSIM ↑ | 0.59 | 0.32 | 0.47 | 0.42 | 0.48 | 0.09 | 0.52 | 0.34 | 0.61 | 0.23 | 0.51 |
> > > | | LPIPS ↓ | 0.53 | 0.60 | 0.60 | 0.57 | 0.55 | 0.76 | 0.54 | 0.59 | 0.50 | 0.72 | 0.56 |
> > > | | PSNR ↑ | 19.70 | 13.15 | 14.58 | 14.49 | 17.22 | 8.66 | 17.24 | 12.24 | 18.41 | 12.60 | 17.31 |
> > > | **Ours** | SSIM ↑ | **0.79** | **0.66** | **0.72** | **0.73** | **0.69** | **0.66** | **0.66** | **0.60** | **0.72** | **0.69** | **0.64** |
> > > | | LPIPS ↓ | **0.39** | **0.41** | **0.43** | **0.39** | **0.44** | **0.45** | **0.49** | **0.54** | **0.45** | **0.50** | **0.51** |
> > > | | PSNR ↑ | **21.73** | **17.83** | **17.75** | **18.52** | **21.60** | **19.51** | **19.03** | **16.48** | **19.50** | **17.10** | **17.63** |

---

### Author Response · Authors · 2025-11-28
**Thank You Note**

### **Dear AC and Reviewers,**

We deeply appreciate the time and effort dedicated to reviewing our work. We have carefully reflected on your feedback, conducted extensive additional experiments, and provided detailed responses throughout the discussion period.We find our previous discussions to be incredibly valuable. Although the discussion phase has been cut short and scores reverted due to the recent policy update, **we want to emphasize that your time and effort were not in vain. The discussion we shared has significantly contributed to improving the quality of our paper.** We truly treasure your insights and sincerely hope this incident does not discourage you from the vital work of peer review. Thank you for your dedication.

Sincerely,
The Authors

---

### Author Response · Authors · 2025-12-03
**Summary of Rebuttal and Revision**

## Dear PC, SAC, AC, and Reviewers,
Thank you again to the reviewers for their thoughtful and constructive feedback, and to the AC for taking on this additional responsibility in light of the recent OpenReview incident. We are delighted to report that we have engaged in a fruitful discussion with all reviewers, and **every reviewer has expressed positive feedback following our rebuttal.** Specifically:

### **Reviewer Conclusions**

* Reviewer S5jM (Originally Score 2): Explicitly stated: "**I will raise my previous rating**... **I would raise my score to accept** if the authors could add the comparison results of traditional SLAM method (even if the proposed method does not outperform them)."

* Reviewer eTqG (Originally Score 6): Explicitly stated: "I am satisfied with the response and **support the paper's acceptance.**"

* Reviewer 91zN (Originally Score 4): Explicitly stated: "My concerns have been sufficiently addressed, and **I will therefore raise my score accordingly.**"

* Reviewer 5jYg (Originally Score 8): Maintained a strong positive rating and explicitly stated: "**My original concerns have been fully addressed**. The paper presents an interesting idea, and the experimental results convincingly demonstrate the effectiveness of the proposed methods."

### **Strengths Highlighted by Reviewers**

**Novelty/Significance:**

* Reviewer eTqG: "The combination of a recurrent feed-forward model with 2DGS... is novel. The use of hidden states... is creative."

* Reviewer 5jYg: "...introduces a novel feed-forward paradigm... motivation is clear and well-founded."

* Reviewer S5jM: "...sophisticated frontend and backend component designs... designs are reasonable and well-supported".

**Performance/Efficiency:**

* Reviewer S5jM: "The experimental results demonstrate the superiority of the proposed Flash-Mono."

* Reviewer eTqG: "...achieves strong results in both tracking (ATE) and rendering (PSNR, SSIM, LPIPS), outperforming recent GS-SLAM systems."

* Reviewer 5jYg: "...demonstrate both the effectiveness and efficiency... significantly improving efficiency."

* Reviewer 91zN: "...offering a plausible path toward faster monocular Gaussian Splatting without fully sacrificing reconstruction quality."

### **Summary of Revisions**

To address the reviewers' constructive feedback, we have made the following improvements:

1. **Generalization:** We added evaluation on the outdoor benchmark KITTI to demonstrate our generalization to outdoor scenes with large scale variance.

2. **Efficiency Analysis:** We added statistics and discussions on model size(Appendix C.1), Gaussian count comparisons(Appendix F), and a detailed runtime breakdown(Appendix E).

3. **Future Direction:** We carried out case study and added a detailed discussion on long-term consistency and lifelong mapping as a prospective future direction (Appendix G).

4. **Baseline Comparisons:** We added comparisons with Photo-SLAM, VGGT-SLAM, and GO-SLAM.

5. **Clarifications:** We clarified  training data usage and numerous technical details regarding the system pipeline and loop closure mechanism.

We are also actively updating the manuscript to incorporate the valuable insights and discussions from the rebuttal phase.

Sincerely,
The Authors

---

### Meta-Review · Area_Chair_GYm2 · 2025-12-09

**Summary:**

Generalization & scope.  Indoor-only evaluation; need outdoor/large-scale or unseen indoor benchmarks (S5jM, eTqG, 91zN, 5jYg)

Baselines & fairness. Add traditional SLAM and recent feed-forward/GS/NeRF SLAM; stronger ablations; clearer novelty boundaries (S5jM, 91zN, 5jYg)

Runtime claims. Clarify “10× speedup,” define end-to-end FPS, and provide per-module runtime under a unified protocol (91zN)

Geometry evaluation. Rendering metrics don’t ensure geometry; request geometric metrics (e.g., completion/chamfer) (S5jM)

Hidden-state mechanism. Analyze loop closure and long-term consistency; discuss prospects for lifelong mapping (eTqG, 91zN)

System details & practicality. Training data, relocalization, loop-correction side effects; memory/model size; robustness under blur/low texture (S5jM, eTqG, 91zN)

Data overlap. Justify CUT3R pretraining on ScanNet v2 vs v1 evaluation (91zN)

**Reviewer Concerns:**

Outdoor/large-scale eval: Addressed. The authors added (KITTI) with ATE + rendering metrics; discussion of scale consistency over long sequences (S5jM, eTqG, 91zN, 5jYg)

Broader baselines: Addressed. Like tracking vs ORB-SLAM3/DROID-SLAM/Photo-SLAM; view-synthesis vs GO-SLAM; a table for VGGT-SLAM (noting semi-online latency); plus existing MonoGS/DepthGS/MASt3R-SLAM (S5jM, 91zN, 5jYg, eTqG)

Runtime clarity: Addressed. FPS defined as end-to-end; detailed per-module timing and loop-closure sparsity; unified timing protocol in appendix (91zN)

Geometry proxy: Addressed. The authors gave a rationale against chamfer/completion for sparse Gaussians; added scale-aligned Depth-L1 table instead (Flash-Mono best) (S5jM)

Hidden-state analysis: Addressed.  ablations vs no loop closure and vs PnP/RANSAC; description of “bag of hidden states” relocalization (eTqG, 91zN)

System details: Addressed. Now available are a training data & curriculum, precise pipeline and interaction frequency, relocalization order, loop-correction trade-offs (temporary quality dip then recovery) (S5jM, 91zN)

Model size & deployability: Addressed. With parameter breakdown; CUDA Graphs + quantization on laptop 4060; feasibility argument for Jetson AGX Orin (eTqG)

Data overlap justification: Addressed. CUT3R init vs v1 eval explained; parity with other methods’ training domains; BundleFusion cited as OOD (91zN)

Deeper novelty isolation: Partially addressed.  Fewer component-swap ablations than requested  (91zN, S5jM)

Lifelong mapping / long-term: Partially addressed. Promising case study + proposed strategies (state replacement, confidence-guided TTT) but no dedicated multi-session experiments yet (eTqG)

Targeted robustness: Addressed.  Motion-blur/low-texture resilience argued via ScanNet + baseline wins (eTqG)

Broader unseen indoor sets: Basically addressed. The reviewers suggested TUM/7-Scenes/ETH3D; the rebuttal added KITTI and relied on BundleFusion OOD, but not those specific sets (91zN)

Failure analysis: Partially addressed. Some discussion (loop-correction artifacts) but no full failure-mode breakdown. (S5jM)

**Reviewer Scores:**

The original scores are 2, 4, 6, 8
One reviewer, who initially gave 4, was convinced by the authors' comments and wanted to upgrade.
Another reviewer, who initially gave 6, was convinced to keep on defending acceptance.

---

### Decision · Program_Chairs · 2026-01-26

Accept (Poster)